# The *Plasmodium* CSP repeats have elastic properties with a critical role in sporozoite motility

Amanda E Balaban[1,8], Sachie Kanatani [1], Jaba Mitra[2,3], Jason Gregory [1], Natasha Vartak[1], Ariadne Sinnis-Bourozikas[1], Fredrich Frischknecht [4,5], Taekjip Ha [2,6,7] & Photini Sinnis [1✉]

## Abstract

The *Plasmodium* circumsporozoite protein (CSP) is the major surface protein of the sporozoite, the infective stage of the malaria parasite. The central repeat region of CSP is the primary immunological target of the sporozoite stage, yet little is known about its structure or function. Here, we show that sporozoite mutants with truncated or scrambled CSP repeats exhibit impaired motility due to altered adhesion site formation and dynamics. Since CSP forms the environment in which the thrombospondin-related anonymous protein (TRAP)-containing adhesion sites assemble, our data suggest that the dense CSP coat is altered in the repeat mutants, affecting adhesion site formation. We hypothesized that this role depends on the biophysical properties of the repeats, and used single-molecule fluorescence-force spectroscopy to test this hypothesis. Our results indicate that the CSP repeats behave like a stiff, linear spring with elastic properties that depend on its length and are lost when the repeats are scrambled. These data provide evidence for a functional role of the CSP repeat region during *Plasmodium* infection and motility.

**Keywords** Malaria; Circumsporozoite Protein; Motility; Repeats; Sporozoite
**Subject Categories** Microbiology, Virology & Host Pathogen Interaction; Structural Biology

## Introduction

Every year, there are 228 million new malaria cases resulting in an estimated 608,000 deaths, the majority of which are in children in Sub-Saharan Africa (World Health Organization, 2023). Malaria is caused by protozoan parasites of the genus *Plasmodium*, whose complex life cycle requires cycling between mosquito and vertebrate hosts. Infection in the vertebrate host is initiated when mosquitoes inoculate sporozoites into the dermis. Sporozoites are actively motile and migrate through the skin to find and enter the blood circulation, which carries them to the liver, where they actively enter hepatocytes and initiate the next life cycle stage. Transmission is a bottleneck for the parasite and, as such, has proven to be a promising point for intervention.

The major surface protein of the sporozoite, the circumsporozoite protein (CSP), forms a dense coat on the parasite's surface. It is attached to the plasma membrane by a glycosylphosphatidylinositol (GPI) anchor (Nagar et al, 2024) and its overall structure is conserved among *Plasmodium* species (Fig. 1A); consisting of a highly conserved adhesive domain, the type I thrombospondin repeat (TSR) in the carboxy-terminus, a central repeat region, and an N-terminus that ends with a conserved proteolytic cleavage site (Region I; (Coppi et al, 2005)). Previous studies have shown that controlled exposure of the TSR domain functions to guide sporozoites as they migrate from the mosquito midgut to the mammalian liver (Coppi et al, 2011). The N-terminal domain masks the TSR as sporozoites migrate from the mosquito midgut to the mammalian liver and once in the liver, a sporozoite protease cleaves CSP at Region I, leading to the removal of the N-terminus and exposure of the TSR, an event that is associated with the switch from a migratory to an invasive sporozoite (Coppi et al, 2007; Coppi et al, 2011). These findings demonstrated that the N- and C-terminal portions of CSP are likely distinct domains and led to structural studies by domain, since full-length CSP could not be crystallized. The crystal structure of the C-terminal TSR and conserved upstream sequence demonstrated some unique structural features compared to canonical TSRs, including a hydrophobic pocket that may be important in host-parasite interactions (Doud et al, 2012). Though the N-terminus has not been crystallized, recent biophysical studies show that it is an intrinsically disordered region that is highly flexible (Geens et al, 2024). Together, these studies have begun to unravel the function of the N- and C-terminal domains of CSP. In contrast, little functional work has been performed on the central repeat region, despite their unique primary sequence and their importance as a target of protective antibodies after vaccination.

The repeat region of CSP consists of tandem amino acid repeats 130 to 190 amino acids in length. Though the sequence varies

[1]Johns Hopkins Malaria Research Institute and Department of Molecular Microbiology & Immunology, Johns Hopkins Bloomberg School of Public Health, Baltimore, MD 21201, USA. [2]Department of Biophysics and Biophysical Chemistry, Johns Hopkins School of Medicine, Baltimore, MD 21205, USA. [3]Department of Materials Science and Engineering, University of Illinois at Urbana-Champaign, Champaign, IL 61801, USA. [4]Integrative Parasitology, Center for Infectious Diseases, Heidelberg University Medical School, Heidelberg, Germany. [5]German Center for Infection Research (DZIF), Heidelberg, Germany. [6]Howard Hughes Medical Institute and Program in Cellular and Molecular Medicine, Boston Children's Hospital, Boston, MA 02115, USA. [7]Department of Pediatrics, Harvard Medical School, Boston, MA 02115, USA. [8]Present address: IDEXX Laboratories, Westbrook, ME, USA. ✉E-mail: psinnis1@jhu.edu

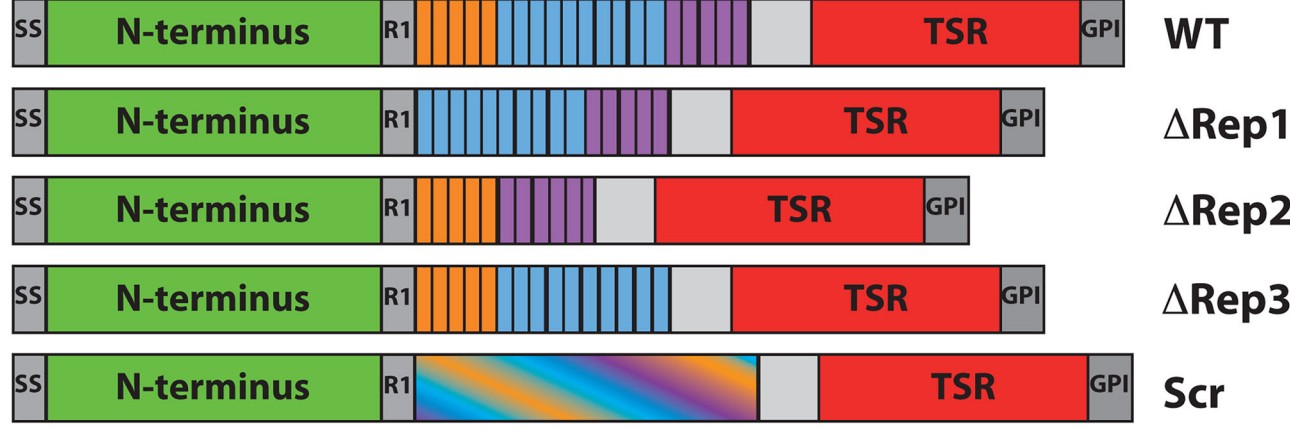

**A**

**WT Repeat Sequence**

KLKQP PPPP NPNDPPPP NPNDPPPP NPNDPPPP NPNDPAPP NANDPAPP NANDPAPP NANDPAPPNANDPAPP NANDPAPP NANDPAPP NANDPPPP NPNDPAPP QGNNN PQPQ PRPQ PQPQ PQPQ PQPQ PQPQ PRPQ PQPQP

**Scr Repeat Sequence**

KLKQP PPPP PNNAPPRD GPPAAPPD PDQNPDPD PQPPQPQN PPNPPPNP NPQDPNNP PQDQPANP NPPPPAQP PPDPDPPA QPNPPANP NNPPDPAP NPNPPQPN QAPNPPQN QPNDPPPA NNDNAQPN PANRNPPP PPAQP

Panels labeled WT, ΔRep1, ΔRep2, ΔRep3, Scr showing protein domain structures (SS, N-terminus, R1, repeat regions, TSR, GPI).

**B**

Western blot lanes: WT-GFP, RCon, ΔRep1, WT-GFP, RCon, ΔRep3, WT-GFP, ΔRep2, WT-GFP, Scr

Molecular weight markers: 100, 50, 37, 25

Anti-TRAP

Anti-CSP (C-Dis)

Salivary Gland | Hemolymph

**C**

Y-axis: SG Sporozoites/Mosquito ($10^2$ to $10^5$)

X-axis groups: Control ΔRep1, Control ΔRep2 (****), Control ΔRep3, Control Scr (****)

**D**

Control | ΔRep2 | Scr

**Figure 1. CSP expression and salivary gland invasion of repeat mutants.**

(A) Schematic of CSP repeat mutants. SS, signal sequence; R1, region I; TSR, type I thrombospondin repeat; GPI, glycosylphosphatidyl inositol anchor. The amino acid sequence above the diagram is the wild-type *Plasmodium berghei* repeat sequence, with the region I cleavage site in gray. The three repeat sequences are color-coded in both the sequence and CSP schematic: orange = first repeat, blue = second repeat, and purple = third repeat. The scrambled repeats are indicated with the blending of the different repeat block colors, and the scrambled sequence is shown below the diagram. (B) CSP expression in repeat mutants. Lysates of salivary gland or hemolymph repeat mutant and control sporozoites were subject to western blot analysis, using antisera specific for the disordered region in the C-terminus of CSP to visualize CSP expression (anti-CSP C-Dis, specific for the sequence CDDSYIPSAEKILEFVKQIRDSITE), with TRAP antisera as a loading control. For ΔRep2 and Scr blots, hemolymph sporozoites were used because of the limited numbers of salivary gland sporozoites. Molecular weight markers are indicated on the left. (C) Salivary gland sporozoite invasion. Salivary glands from 20 infected mosquitoes were pooled, homogenized, and the average number of sporozoites per mosquito was determined using a hemocytometer. Each data point is from an independent mosquito cycle with matched control and mutant cycles. Kruskal–Wallis $P < 0.0001$. Mann–Whitney–Wilcoxon ****$P < 0.0001$. (D) Live confocal imaging of infected mosquito salivary glands (outlined in gray) of Control, ΔRep2, and Scr-infected mosquitoes with GFP channel (sporozoites) and DIC. GFP channel is pseudo-colored with different colors representing different positions in Z, allowing for visualization of sporozoite distribution through the depth of salivary gland tissue. Scale bars = 25 μm. Color bar scales on top right indicate section depth. Source data are available online for this figure.

among *Plasmodium* species (Appendix Table S1; plasmodb.org/plasmo/app), they share several features, suggesting a conserved structure and/or function across species. These features include: (1) repeat sequences are composed of a subset of amino acids, with 80% of the residues being some combination of proline, glycine, alanine, and asparagine; (2) an isoelectric point between 2.9 and 3.9; and (3) repeat sequences of different *Plasmodium* species are exchangeable, with no observable phenotype (Persson et al, 2002; Espinosa et al, 2013). Previous studies on the structure of the CSP repeats have used repeat peptides or recombinant protein and provided evidence for a β-turn conformation (Dyson et al, 1990; Fasman et al, 1990; Verdini et al, 1991; Ghasparian et al, 2006; Plassmeyer et al, 2009). In addition, studies with recombinant CSP bound to repeat-specific antibodies showed that the repeats can take on a spiral structure, though this may be induced or stabilized by interaction with an antibody (Oyen et al, 2017). Because it has not been possible to conduct studies on native protein and no assays exist to confirm proper folding, the structure of native CSP on the sporozoite surface is not known. Functional studies of the repeats have not been performed, and hypotheses as to their function have ranged from a disordered linker region to an immune screen. Since high titers of anti-repeat antibodies immobilize sporozoites, the possibility that they have a role in motility has been proposed. To probe the function of the CSP repeat region, we generated mutant parasites with truncated or altered repeats in the rodent malaria model *Plasmodium berghei* and interrogated the biophysical properties of the repeats using single-molecule fluorescence-force spectroscopy.

## Results

### Generation of CSP repeat region mutants

Previous work has shown that deletion of the entire CSP repeat region results in severe defects in sporozoite development, with decreased oocyst sporozoite numbers and sporozoite death prior to exiting the oocyst (Ferguson et al, 2014). Since none of these sporozoites reached the salivary glands, it was not possible to elucidate the function of the repeats in the mammalian host using this mutant. To overcome this limitation, we created mutants with more subtle phenotypes. We previously established that during migration within the mosquito, the exposure of the N- and C- terminal domains of CSP on the surface is altered to modify sporozoite adhesion (Coppi

et al, 2011). To probe whether the length of the repeats was important for domain exposure, we generated three CSP repeat truncation mutants, ΔRep1, ΔRep2, and ΔRep3, in which different lengths or regions of the repeats were deleted (Fig. 1A). In addition, to test whether the repetitive nature of the repeats contains important structural features we also generated a scrambled mutant, Scr, by randomly scrambling the amino acid sequence of the repeats. For this mutant, we avoided introduction of new secondary structures, as predicted by the Chou-Fasman method, but maintained amino acid content and length (Fig. 1A). These mutants were generated in *P. berghei* ANKA 507 clone 1, a line that constitutively expresses GFP and was selected by flow cytometry for the removal of the selection cassette (Janse et al, 2006a). Transfection plasmids containing the desired *csp* repeat mutations were designed to replace the endogenous wild-type *csp* gene (Appendix Fig. S1A). Transfected parasites were cloned and verified by a series of diagnostic PCRs and sequencing of the *csp* gene (Appendix Fig. S1B). Phenotyping of CSP repeat mutants was always performed in parallel with a control. In some cases, this was the parental line *P. berghei* ANKA 507 clone 1 (WT-GFP) and in other cases, this was a recombinant control line (RCon) in which a wild-type *csp* was transfected into the native locus, controlling for any effects that the altered genomic locus might have. Side-by-side comparisons of WT-GFP and RCon parasites showed that they had similar phenotypes in the assays used in our study, though RCon sporozoites had slightly reduced maximum speed and time with multiple adhesion sites (Appendix Fig. S2). Thus, throughout the manuscript, "Controls" refer to either WT-GFP, RCon or a combination of both lines.

### ΔRep2 and Scr sporozoites do not progress normally through the mosquito

To characterize the progression of CSP repeat mutants through the mosquito stages, we first assessed the number of salivary gland sporozoites in each line. As shown in Fig. 1C, ΔRep1 and ΔRep3 had comparable salivary gland infections to controls. In contrast, ΔRep2 and Scr parasites were inhibited in their capacity to enter the salivary glands, with fewer salivary gland-associated sporozoites. We confirmed that ΔRep2 and Scr salivary gland-associated sporozoites were inside the salivary glands, as opposed to attached to the external surface of the glands, by live confocal imaging of salivary glands (Fig. 1D).

Since ΔRep2 and Scr exhibited lower numbers of salivary gland sporozoites, we characterized their mosquito phenotypes in more

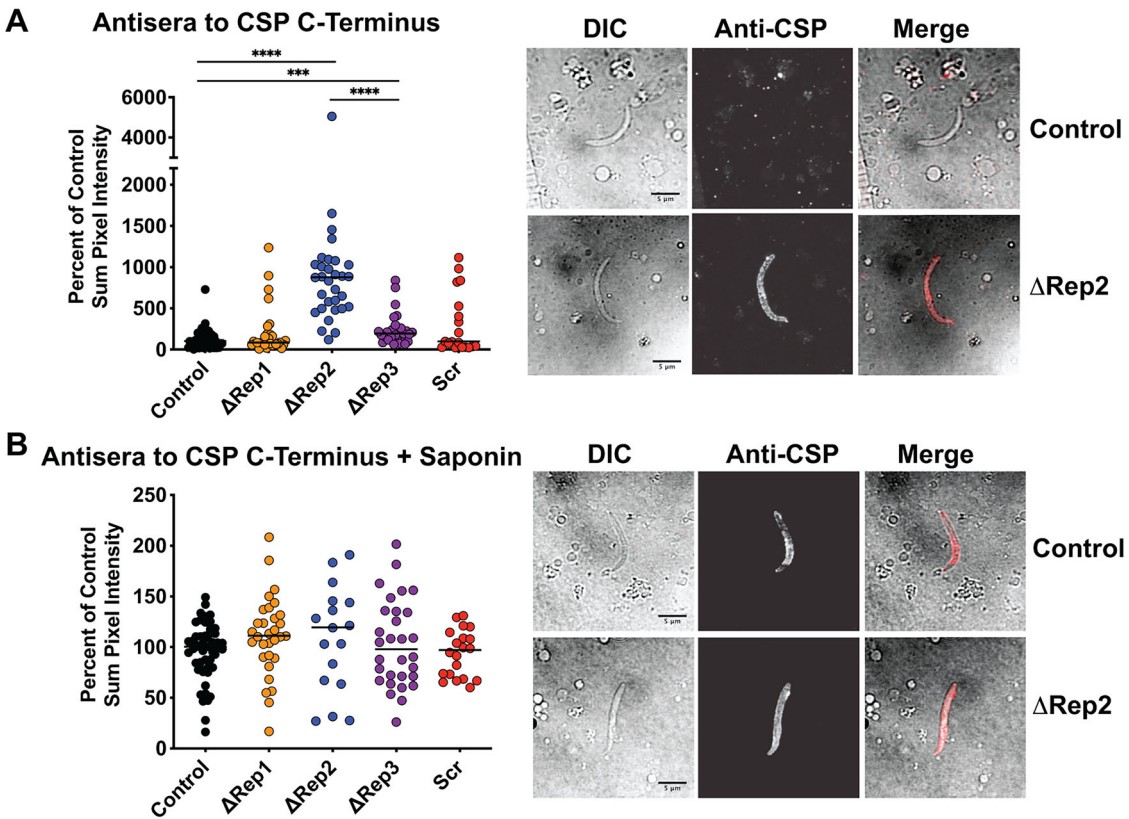

**Figure 2.  ΔRep2 sporozoites have enhanced exposure of the carboxy-terminal TSR.**

Salivary gland sporozoites of the indicated mutant lines were stained with antibodies specific to the C-terminal TSR domain of CSP. Sporozoites were unpermeabilized (**A**) or, 0.1% saponin was included with the antisera (**B**). In total, 20–30 sporozoites per condition were quantified for the sum pixel intensity and presented as a percentage of the median pixel intensity of matched control salivary gland sporozoites. For (**A**), Kruskal–Wallis $P < 0.0001$ followed by Mann–Whitney–Wilcoxon ***$P = 0.0002$, ****$P < 0.0001$. For (**B**) Kruskal–Wallis $P = 0.1980$. Representative images of Control and ΔRep2 for each antibody condition are shown to the right of each panel with DIC, fluorescence and merged images. Scale bars = 5 μm. Source data are available online for this figure.

detail. ΔRep2 and Scr sporozoite development in oocysts was unaltered compared to controls, with a comparable number of midgut sporozoites per mosquito in five independent mosquito cycles (Appendix Fig. S3). We then investigated whether ΔRep2 and Scr mutants had defects in egress from oocysts by quantifying sporozoite numbers in the hemolymph, the open circulatory system of the mosquito. We found that these mutants had similar numbers of hemolymph sporozoites as controls, indicating that they exit oocysts normally (Appendix Fig. S3). These data confirm that the decreased number of salivary gland sporozoites in ΔRep2 and Scr mutants is due to a salivary gland invasion defect and not an earlier developmental phenotype.

## Expression and conformation of mutant CSP

To determine whether CSP was expressed at similar levels in the repeat mutants as in control sporozoites, we performed western blot analysis with antisera generated to the disordered region just after the repeats (anti-CSP C-Dis), since we could not use the CSP repeat antisera with our mutants. By western blot CSP is generally observed as two bands, a full-length form and a cleaved form (Yoshida et al, 1981; Cochrane et al, 1982; Coppi et al, 2005). As shown in Fig. 1B, mutant sporozoites expressed similar amounts of

CSP, demonstrating that the repeat mutations did not alter CSP expression. To perform these experiments with the ΔRep2 and Scr mutants, we used hemolymph sporozoites, as the low number of salivary gland sporozoites resulted in large amounts of contaminating mosquito debris, which gave a high background on western blots. Of note, these were the only experiments in this study for which we used hemolymph sporozoites.

We then went on to determine whether mutant CSP was conformationally altered, using previously developed polyclonal antisera specific for the C-terminal TSR domain of CSP (Coppi et al, 2005). Previous studies found that the C-terminal adhesion domain was exposed only during sporozoite development in the oocysts and upon reaching the liver (Coppi et al, 2011). In between these two host locations, the N-terminal domain masks the C-terminal TSR domain. By immunofluorescence microscopy, we found that salivary gland sporozoites of ΔRep1 and Scr CSP repeat mutants did not stain with the C-terminal antisera, similar to controls, while ΔRep3 sporozoites had a mild increase in TSR exposure and ΔRep2 sporozoites had significantly increased TSR exposure compared to both control and ΔRep3 sporozoites (Fig. 2A). Treatment of sporozoites with 0.1% saponin during staining with the anti-C-terminal antibody to break intramolecular interactions in the surface CSP and expose the C-terminal domain,

allowed for the comparison of total CSP on the surface of mutant sporozoites. Using this method, we found comparable levels of CSP on the surface of control and mutant sporozoites (Fig. 2B), confirming our western blot results.

The enhanced C-terminal exposure on ΔRep2 salivary gland sporozoites could be explained by one of two mechanisms: (1) an inability of the N-terminus to mask the C-terminus in this mutant due to the shortened repeat linker or (2) premature proteolytic processing of the N-terminus of CSP, which would lead to exposure of the C-terminal domain (Coppi et al, 2005; Coppi et al, 2011). To determine if premature cleavage of the N-terminus could account for the exposure of the C-terminus in ΔRep2 sporozoites, we metabolically labeled CSP in control, ΔRep1 (as an additional control), and ΔRep2 salivary gland sporozoites to observe CSP cleavage. No differences in CSP cleavage were observed in ΔRep2 sporozoites (Appendix Fig. S4), suggesting that CSP is not prematurely cleaved. More definitive proof would be a pulse-chase experiment, but we could never harvest sufficient numbers of ΔRep2 sporozoites to perform this experiment. Thus, current evidence suggests, but does not definitively prove, that the exposure of the TSR in ΔRep2 sporozoites is not due to premature CSP cleavage but rather to insufficient length of the mutant repeats to allow for N-terminal domain masking of the TSR.

## ΔRep2 and Scr sporozoites have decreased infectivity for hepatocytes

To assess the capacity of CSP repeat mutant salivary gland sporozoites to infect mice, we intravenously inoculated sporozoites into mice and quantified relative parasite abundance in the liver using RT-qPCR (Bruna-Romero et al, 2001). While ΔRep1 and ΔRep3 sporozoites had similar liver stage parasite burdens as controls, infection with ΔRep2 and Scr salivary gland sporozoites resulted in significantly lower liver parasite burden (Fig. 3A). To determine whether the defects in ΔRep2 and Scr infectivity were due to defects in invasion or development, we performed in vitro hepatocyte infection assays with Hepa1-6 cells and Scr sporozoites, as ΔRep2 salivary gland sporozoite yields were too low for these assays. Infection of hepatocytes with the same number of control or Scr sporozoites resulted in fewer Scr exoerythrocytic forms (EEFs) compared to controls (Fig. 3B). However, Scr EEFs did not have defects in development, as determined by EEF size (Fig. 3C), suggesting that Scr sporozoites are defective in entering hepatocytes but once inside, they develop normally. Defective infection of hepatocytes by both ΔRep2 and Scr sporozoites was sufficient to delay the initiation of blood stage infection (pre-patency period) (Fig. 3D).

## ΔRep2 and Scr sporozoites have gliding motility deficits

ΔRep2 and Scr sporozoites exhibited a diminished capacity to infect two very different tissues: the salivary glands of mosquitoes and the liver of mice. Lack of tissue specificity in the infectivity phenotype of the CSP repeat mutants suggests these phenotypes are not the result of interrupted receptor-ligand interactions, but due to a feature required of sporozoites at both stages. Sporozoites move by a substrate-based motility called gliding motility that is powered by a subpellicular actin-myosin motor (Baum et al, 2006). Previous studies have demonstrated that sporozoites with motility defects

cannot invade salivary glands or hepatocytes efficiently (Sultan et al, 1997; Ejigiri et al, 2012). Therefore, we analyzed the motility phenotypes of CSP repeat mutant salivary gland sporozoites using several in vitro assays.

We began with live gliding assays, recording salivary gland sporozoite motility on glass coverslips for 2.5 min. Approximately 90% of sporozoites were attached and motile across the studied lines (Appendix Fig. S5). We categorized sporozoite motility according to previously observed motility phenotypes: continuous circular gliding, patch gliding, attached waving, or meandering (Vanderberg et al, 1974; Stewart et al, 1988; Hegge et al, 2010). Examples of circular gliding, patch gliding, and attached waving are shown in Fig. 4A and in Movies EV1, EV2, and EV3. Circular gliding, in which sporozoites move in continuous circles, is the predominant form of motility observed in wild-type parasites and is an indicator of fully mature and infectious sporozoites (Movie EV1). Meandering is a form of circular gliding in which the sporozoites move in longer, more open arcs rather than tight circles (Stewart et al, 1988). Patch gliding sporozoites move back and forth over one area and do not demonstrate continuous forward motility. Patch gliding is rarely observed in wild-type salivary gland sporozoites, though it is frequently observed in midgut sporozoites and in mutants such as the TRAP KO parasite, where parasite adhesion is defective (Sultan et al, 1997; Kappe et al, 1999) (Movie EV2). These data have led to the hypothesis that patch gliding is a step in the maturation of the motility machinery (Münter et al, 2009). Waving parasites are attached at one end and flex the other end (Vanderberg et al, 1974; Hegge et al, 2010) (Movie EV3). We found that like controls, the majority of ΔRep1 and ΔRep3 sporozoites engaged in circular gliding, with a small minority patch gliding or waving (Fig. 4B). In contrast, ΔRep2 and Scr sporozoites had a significantly lower proportion of sporozoites engaged in circular gliding (12% ΔRep2 and 27% Scr), while the majority of these mutants were either patch gliding, (39% ΔRep2 and 46% Scr) or waving (41% ΔRep2 and 25% Scr) (Fig. 4B). These differences in motility phenotypes between the ΔRep2 and Scr sporozoites and controls were statistically significant (Z-test for proportions, $P < 0.00001$).

## ΔRep2 and Scr sporozoites are slower circular gliders, but faster patch gliders

Substrate-based gliding motility of sporozoites requires the cyclic formation and dissolution of adhesion sites (Münter et al, 2009; Hegge et al, 2010; Ejigiri et al, 2012). Adhesion cycles are comprised of the slow development of adhesion sites followed by fast movement over these adhesions (Münter et al, 2009) and can be observed in the instantaneous speed plots of gliding sporozoites (Appendix Fig. S6). To better understand the motility defects in ΔRep2 and Scr sporozoites, we analyzed these instantaneous speed plots to determine how much time each mutant spent at a given speed by combining the data from all movies, extrapolating the speed between every frame of each movie and plotting the frequency of speed distributions. When data from circular gliding and patch gliding sporozoites were combined, ΔRep2 and Scr sporozoites were observed to move less often and at slower speeds (Fig. 5A).

We also separately analyzed the instantaneous speed plots of circular and patch gliding parasites and found that circular gliding

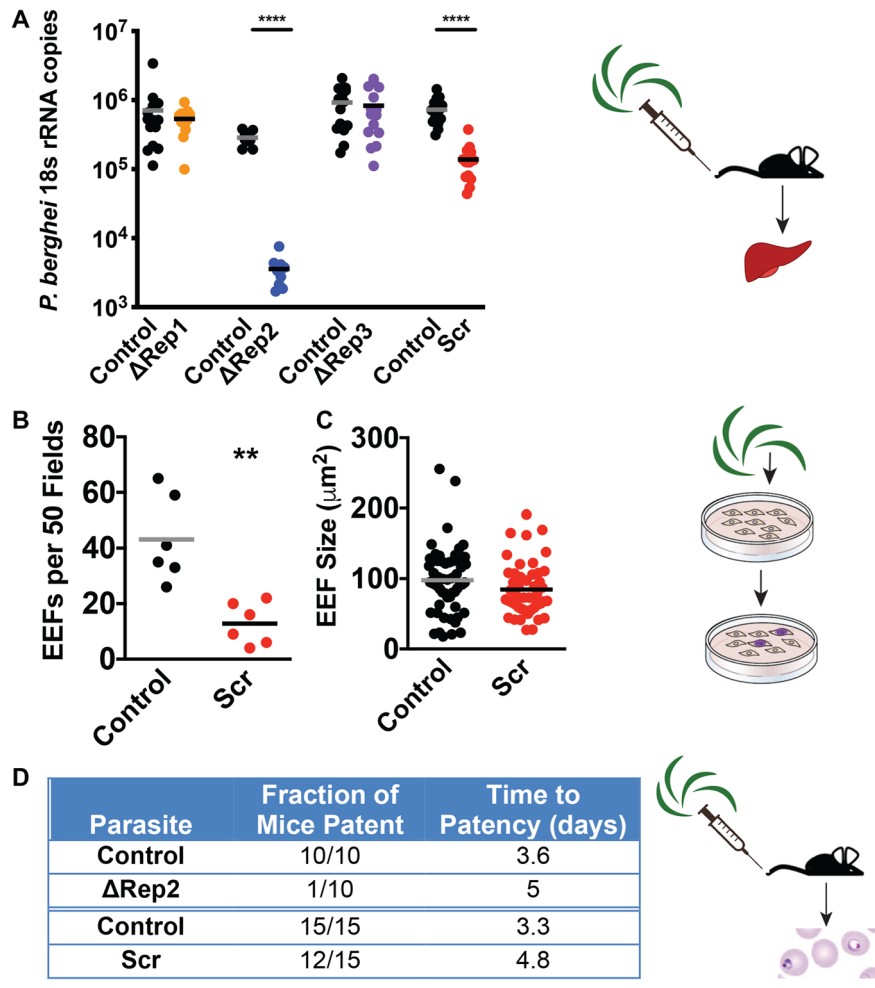

**Figure 3.  ΔRep2 and Scr sporozoites have decreased infectivity in the mammalian host.**

(A) Sporozoite infection assays in mice. 5000 control or CSP repeat mutant sporozoites were inoculated intravenously into C57Bl/6 mice, and 40 h later, parasite liver burden was determined by RT-qPCR using primers specific *P. berghei* 18s RNA. Shown are pooled data from two to three experiments with five mice per group in each experiment. Kruskal–Wallis $P < 0.0001$ followed Mann–Whitney–Wilcoxon ****$P < 0.0001$. (B, C) Development of exoerythrocytic stages (EEFs) in vitro. Control or Scr mutant sporozoites were added to Hepa1-6 cells and 24 h later EEFs were fixed and stained with a-HSP70. Shown are the numbers of EEFs per 50 fields (B) and EEF size at 48 h post-infection (C) from two independent experiments with 3 replicates per experiment. Welch's *t* test, **$P < 0.0032$. (D) Initiation of blood stage infection by ΔRep2 and Scr mutant sporozoites. In all, 5000 sporozoites were inoculated intravenously into mice and mice were monitored by blood smear from day 3 to day 9 post-infection. Shown are the number of mice that became patent and the average day of patency for mice with blood stage infections. Data are pooled from two to three experiments. Source data are available online for this figure.

ΔRep2 and Scr sporozoites spent more time not moving or moving at ≤0.5 µm/s (Fig. 5B) compared to controls or ΔRep1 and ΔRep3 sporozoites. Since circular gliding ΔRep2 and Scr sporozoites spent more time moving at slower speeds, it would be expected that their median speed, over the entire recorded movie, would be slower, and indeed that is what we observed (Fig. 5D, left panel). Nonetheless, no severe defects in maximum speed were observed (Fig. 5D, right panel), suggesting that the motor is still functional in these mutants. In contrast to the circular gliding sporozoites, instantaneous speed distributions of patch gliding ΔRep2 and Scr parasites showed that these mutants exhibited faster speeds than control or ΔRep1 and ΔRep3 patch gliders and spent more time moving (Fig. 5C). These data are consistent with the increased median and maximum speed of ΔRep2 and Scr patch gliding parasites, both of which were higher than controls, ΔRep1 and ΔRep3 sporozoites

(Fig. 5E), further suggesting that the internal motility machinery is intact.

## Adhesion site dynamics in ΔRep2 and Scr sporozoites

To better understand the gliding defects in our CSP repeat mutants, we used reflection interference contrast microscopy (RICM) to observe adhesion site assembly and disassembly over time. In RICM, the light reflected closest to the coverslip is phase-shifted such that cell membranes closely apposed to the coverslip appear dark and can be distinguished from the rest of the cell (Weber et al, 2003). A previous study of circular gliding sporozoites found that an adhesion site first forms at the anterior end of the sporozoite, grows (assembles), sometimes becoming as large as the full length of the sporozoite, and finally disassembles, becoming smaller at the

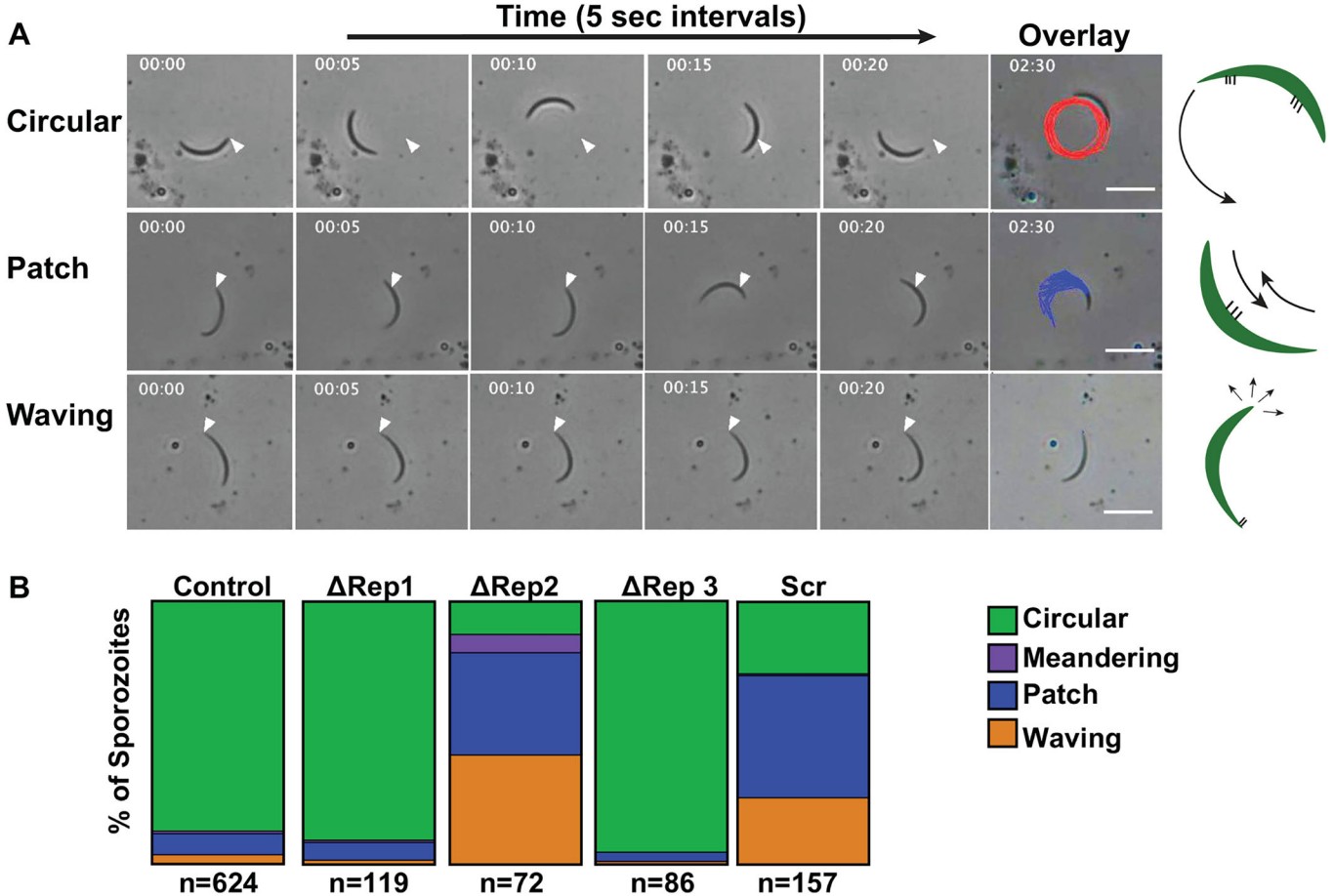

**Figure 4. ΔRep2 and Scr sporozoites exhibit decreased productive gliding motility.**

CSP repeat mutants and control sporozoites were added to glass coverslips, and their motility was recorded in 2 min movies and analyzed. (A) Examples of the three predominant classes of motility behavior observed: Shown are stills from movies in which sporozoites exhibit circular gliding (top), patch gliding (middle), and waving (bottom) with the tracking overlay from the entire movie in the final column. White arrow indicates the position of the sporozoite's posterior end at the beginning of the movie. Scale bars = 10 μm. Cartoons on the right-hand side show sporozoites in green, adhesion sites in thick black lines, and direction of motility (arrows). Meandering is not shown as it is similar to circular gliding but with longer more open trajectories. (B) Quantification of motility types for each mutant. The percent of bar occupied by a given motility behavior corresponds to the percent of sporozoites exhibiting that behavior. Assays with mutant sporozoites were always done side-by-side with controls. At least three experiments per mutant were done. The total number of each genotype analyzed is indicated below each bar. There were statistically significant differences between the proportion of SCR and ΔRep2 sporozoites exhibiting circular gliding, patch gliding, and waving when compared to control sporozoites (Z-test for proportions, $P < 0.00001$). There was also a significant difference in the proportion of meandering sporozoites between ΔRep2 and control sporozoites (Z-test for proportions, $P < 0.001$). Source data are available online for this figure.

posterior end of the sporozoite while the formation of another adhesion site at the anterior end of the sporozoite occurs, thus continuing the cycle (Münter et al, 2009) (Fig. 6A; Movie EV4). In contrast, when patch gliding sporozoites are imaged by RICM, one observes a small site of adhesion over which the sporozoite repeatedly moves back and forth, unable to grow this adhesion site or to form a second adhesion site (Fig. 6A; Movie EV5). Thus, the high proportion of patch gliding ΔRep2 and Scr salivary gland sporozoites suggests that these mutants have altered adhesion site dynamics.

We also used RICM to visualize adhesion site turnover in the small percentage of ΔRep2 and Scr mutants that were engaged in circular gliding. We defined an adhesion turnover event as an adhesion site forming at the anterior end of the sporozoite, translocating to the posterior end, and finally disassembling

(Fig. 6A). When we quantified these events, we found fewer adhesion site turnover events per unit time in ΔRep2 and Scr mutants compared to controls (Fig. 6B). We then went on to dissect adhesion site assembly and disassembly. By observing RICM movies frame-by-frame, we binned each time point as having multiple or single adhesion sites per sporozoite. Control sporozoites spend ~70% of gliding time with multiple adhesion sites (Fig. 6C). In contrast, circular gliding ΔRep2 and Scr sporozoites spend less time with multiple adhesion sites compared to controls (Fig. 6C), suggesting an adhesion site dynamics defect.

## TRAP secretion does not explain adhesion site defects

TRAP (thombospondin-related anonymous protein) is a trans-membrane protein that is critical for gliding motility, functioning

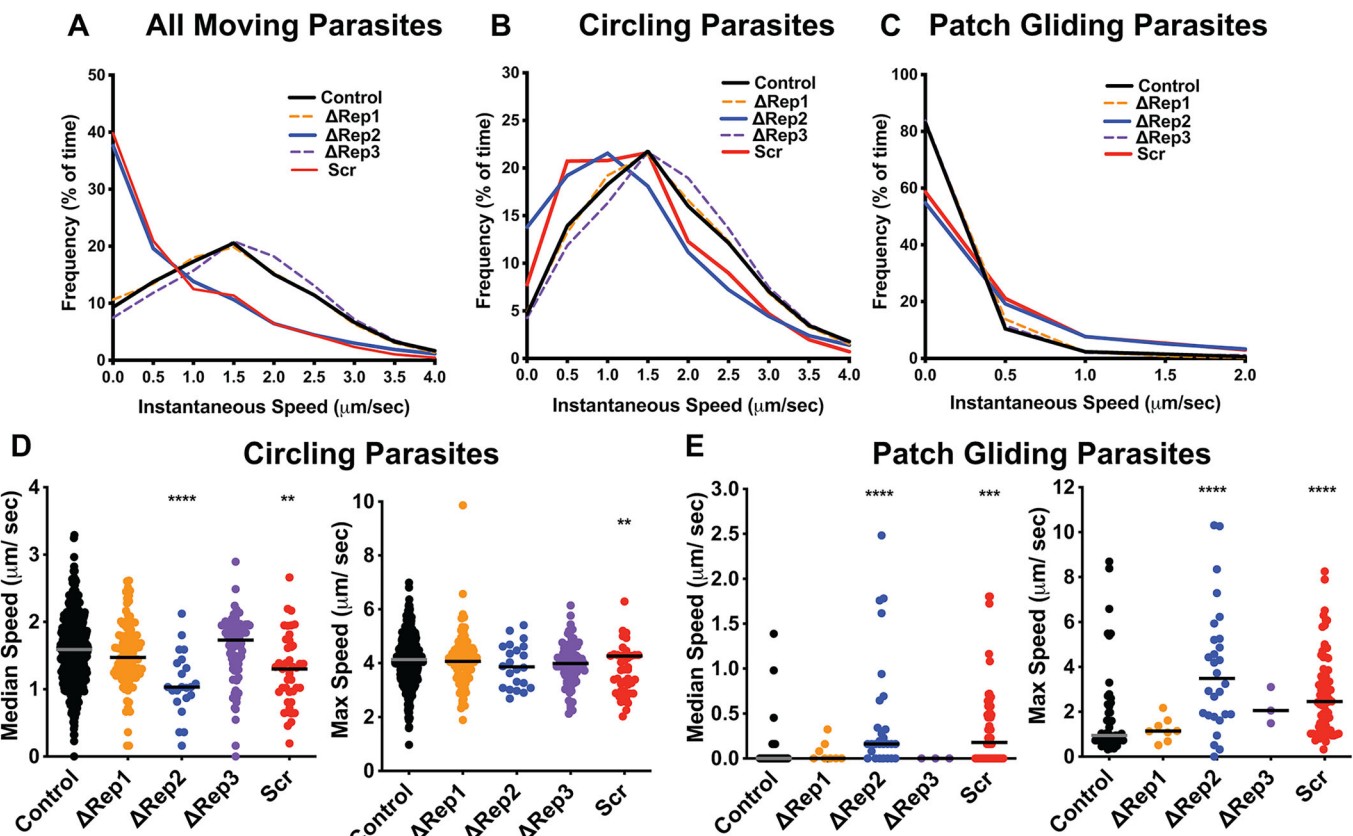

**Figure 5. Instantaneous speed measurements demonstrate that ΔRep2 and Scr have decreased speed during circular gliding but faster patch gliding speeds.**

CSP repeat mutants and control sporozoites were added to glass coverslips, and their motility was recorded in 2.5 min movies. (A–C) Frequency distributions of instantaneous speeds recorded for mutant and control sporozoites that were either circling or patch gliding (A), only circling (B) and only patch gliding (C). To generate these frequency distributions, instantaneous speed plots were pooled to determine the percent of total time that each population of sporozoites spent in each 0.5 μm per second speed bin. (D, E) Graphs show median and maximum speeds for each sporozoite recorded for circling sporozoites (D) and patch gliding sporozoites (E). Each mutant was analyzed from at least three independent mosquito cycles, and at least 20 movies were collected per experiment. Controls are pooled from all experiments. For median speed of circling parasites: One-way ANOVA $P < 0.0001$ followed by Dunnett's test ****$P < 0.0001$, **$P = 0.0013$. For all other datasets, Kruskal–Wallis $P < 0.0001$ followed by Mann–Whitney–Wilcoxon **$P = 0.0043$,***$P = 0.0008$, ****$P < 0.0001$. The number of circular gliding sporozoites analyzed for each parasite line is as follows: Control = 559, ΔRep1 = 108, ΔRep2 = 22, ΔRep3 = 82, Scr = 43. Number of patch gliding sporozoites analyzed for each parasite line are as follows: Control = 50, ΔRep1 = 8, ΔRep2 = 28, ΔRep3 = 3, Scr = 73. Source data are available online for this figure.

to link the subpellicular actin-myosin motor to the extracellular substrate (Fig. 7A) (Sultan et al, 1997; Buscaglia et al, 2003). Previous immunofluorescence assays of permeabilized sporozoites have shown that TRAP is stored in specialized secretory vesicles called micronemes in a nuclear-sparring pattern (Wengelnik et al, 1999; Gantt et al, 2000) and that when stimulated to start moving, it is secreted onto the sporozoite surface and shed by the action of a parasite rhomboid protease into the trails (Gantt et al, 2000; Ejigiri et al, 2012). In mutants that are unable to move, secretion still occurs, but shedding into trails does not, such that significantly more TRAP is observed on the sporozoite surface (Ejigiri et al, 2012). Though the rhomboid cleavage site on TRAP has not been precisely determined, it has been localized to the TRAP transmembrane domain (Ejigiri et al, 2012). Because of TRAP's critical role in motility, we investigated whether it was secreted and distributed on the surface of ΔRep2 and Scr repeat mutants by performing immunofluorescence assays on unpermeabilized gliding sporozoites using polyclonal antisera specific for TRAP (Fig. 7B). Control sporozoites had relatively little TRAP staining on their surface, but

substantial TRAP staining in the trails that gliding sporozoites leave behind (representative image in Fig. 7B and quantification of surface TRAP and TRAP trails in 7C&D). In contrast, we found that ΔRep2 and Scr sporozoites had a marked increase in detectable TRAP on their surface, indicating that it is secreted onto the surface, but TRAP was not detected in trails, suggesting it is not able to be shed (representative images shown in Fig. 7B with quantification of surface TRAP and TRAP trails in 7C&D). While 70–90% of Scr and ΔRep2 sporozoites do not engage in circular gliding (Fig. 4), we would expect to see TRAP trails associated with the small percentage of mutant sporozoites that do engage in circular gliding. However, after counting 200–300 sporozoites, we did not observe any TRAP trails in these mutants (Fig. 7D). These data suggest that while TRAP is secreted onto the sporozoite surface in ΔRep2 and Scr sporozoites, it is not shed from the sporozoite surface, even by the small number of mutant sporozoites that are motile.

The lack of TRAP in trails of the ΔRep2 and Scr repeat mutants, together with defects in adhesion site formation and turnover

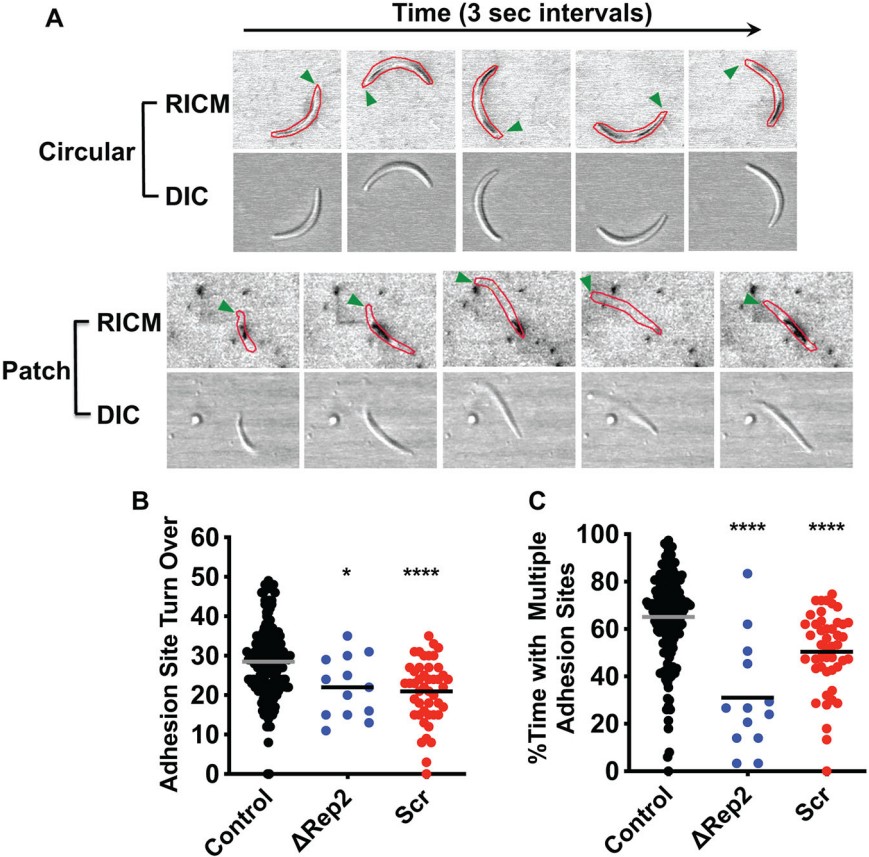

**Figure 6. ΔRep2 and Scr sporozoites have defects in adhesion site formation and turnover.**

Sporozoites were added to glass coverslips and imaged by reflection interference contrast microscopy (RICM) to record adhesion site formation and turnover. (**A**) Sequential stills from RICM imaging of patch gliding and circular gliding sporozoites illustrate adhesion site dynamics during each motility type. RICM images below show adhesion sites with sporozoites outlined in red and green arrowheads indicating the same end of patch gliders and the anterior end of circular gliders. Corresponding DIC images of sporozoites are shown below. (**B, C**) Adhesion site formation and turnover of circular gliding control, ΔRep2, and Scr sporozoites. RICM movies of sporozoites were analyzed to quantify the frequency with which adhesion sites move from the sporozoite's anterior to posterior end in 2.5 min of circular gliding (**B**) and the frequency with which gliding sporozoites have multiple adhesion sites (**C**). One-way ANOVA $P < 0.0001$ followed by Dunnett's test *$P = 0.0126$, ****$P < 0.0001$. Data are pooled from three or more independent experiments with the following number of parasites for each line: Control $= 178$, ΔRep2 $= 13$, Scr $= 46$. Source data are available online for this figure.

demonstrated by RICM analysis, prompted us to visualize adhesion sites in these mutants by staining for myosin tail domain interacting protein (MTIP) in gliding sporozoites. Previous studies have shown that gliding sporozoites have patches of MTIP on their surface that likely represent adhesion sites (Siden-Kiamos et al, 2020; Swearingen et al, 2016). As shown in Fig. 7E, control sporozoites show patches of MTIP on their surface, which can be found on one end, in the middle, or at both ends of the sporozoite, similar to previously published results (Siden-Kiamos et al, 2020; Swearingen et al, 2016). In contrast, ~25% of ΔRep2 and 40% of Scr mutants had only one MTIP dot on their surface, while the remainder had significantly smaller MTIP patches compared to controls (Fig. 7F,G). The abnormal surface staining of both TRAP and MTIP on gliding ΔRep2 and Scr repeat mutant sporozoites, together with their inability to form and rapidly turnover adhesion sites, suggests the underlying problem in these mutants is the organization of protein(s) on the sporozoite's surface. Based on these data, we hypothesize that the sporozoite's dense CSP coat is the environment into which adhesion sites are secreted, with the

repeats possibly forming a higher-order structure that allows for cohesion of the secreted adhesion site, rather than its dispersion in a 'sea' of CSP, a process that is defective when the CSP repeats are scrambled or severely truncated.

## The CSP repeats have elastic properties

These findings raise the possibility that the repeats have conserved biophysical properties, though to date the CSP repeats have resisted structural analysis likely due to their flexible and dynamic nature. Interestingly, CSP repeats have features in common with elastomeric proteins such as spider silk, titin, and abductin, which can crosslink to form a network (Tatham et al, 2000). The elasticity of these proteins is defined by their ability to unfold and fold without energy loss or protein rupture (Tatham et al, 2000). In addition, elastic proteins are difficult to crystallize due to flexible repeats that are high in proline and glycine content and frequently have a ß-turn conformation, which are also features of the CSP repeats.

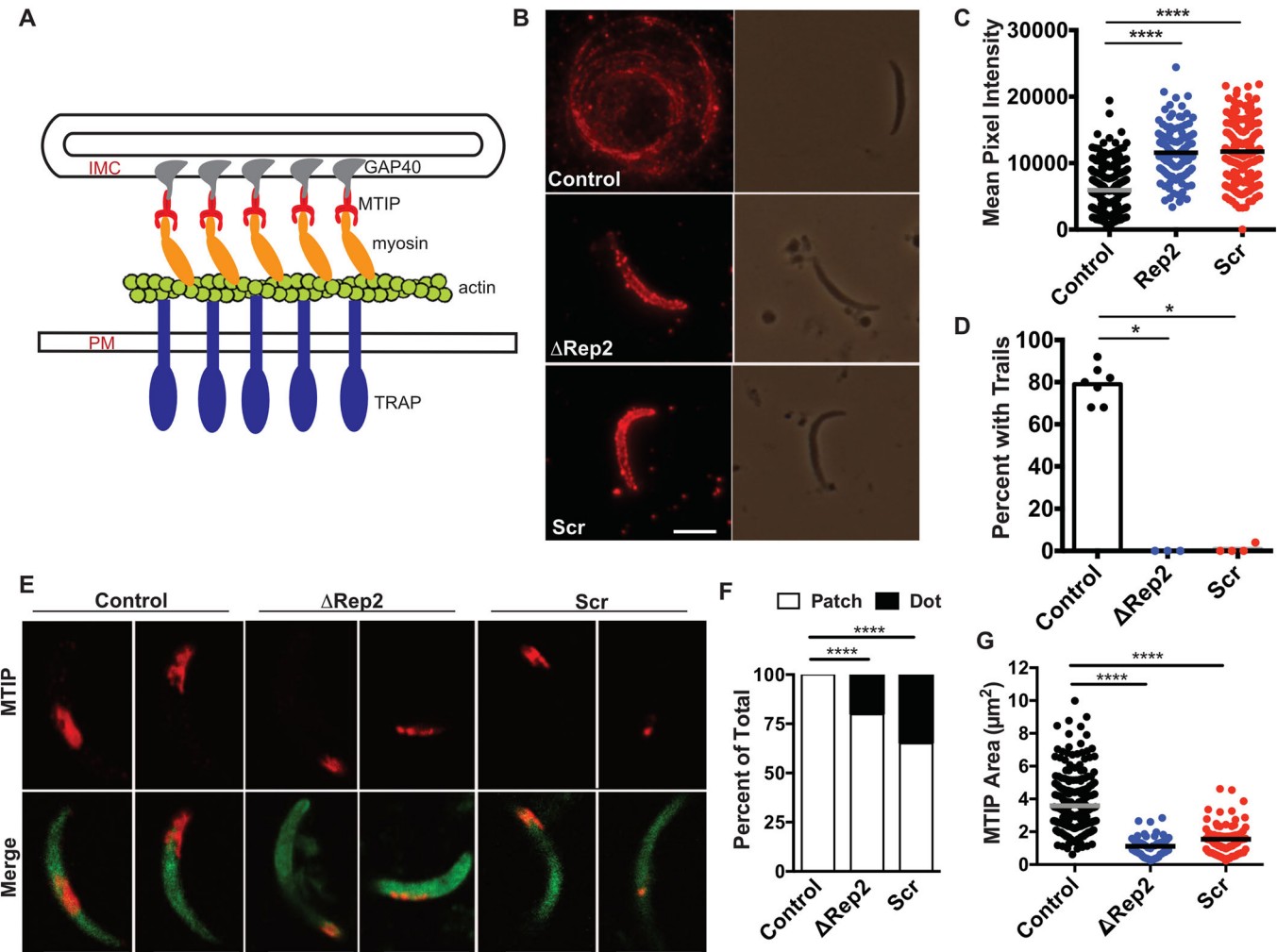

**Figure 7. ΔRep2 and Scr sporozoites have defects in TRAP shedding and make smaller adhesion sites as indicated by MTIP staining.**

(A) Cartoon of the motor showing TRAP in the sporozoite plasma membrane (PM) with its cytoplasmic tail linked to the actin-myosin motor, which in turn is anchored to the inner membrane complex (IMC) via MTIP (myosin tail interacting protein) and various gliding-associated proteins (GAPs), of which GAP40 is shown. (B–D) Surface TRAP on circular gliding sporozoites was assessed by allowing them to glide on glass coverslips for an hour, followed by staining without permeabilization. (B) Representative fluorescent and phase-contrast images of Control, ΔRep2 and Scr sporozoites. Scale bar = 5 μm. (C) TRAP staining on the sporozoite surface was quantified by measuring mean pixel intensity of each sporozoite. One-way ANOVA $P < 0.0001$ followed by Dunnett's test ****$P < 0.0001$. (D) Sporozoites with and without TRAP deposition in trails were counted and expressed as percent of total sporozoites. Data are pooled from three to four independent mosquito cycles with each mutant cycle having a paired control cycle. Each dot represents an experiment with $n$ = 20–100 sporozoites per experiment. Kruskal–Wallis $P = 0.0005$ followed by Mann–Whitney–Wilcoxon *$P = 0.0169$ (control vs ΔRep2), *$P = 0.0136$ (control vs Scr). (E–G) MTIP staining on circular gliding sporozoites was assessed by allowing sporozoites to glide on glass coverslips, followed by fixation with paraformaldehyde and staining with antibodies specific for MTIP. Representative images of MTIP staining are shown in (E). Scale bar = 5 μm. Quantification of staining pattern in which a "dot" is circular whereas a "patch" is more rectangular and of varying size is shown in (F). The area occupied by MTIP dots and patches for individual control and mutant sporozoites is shown in (G). In total, 20–50 sporozoites per line were identified by endogenous GFP and following this MTIP staining, and the area occupied by MTIP was quantified using Fiji. The staining pattern was manually counted with an example of the "dot" pattern shown in the second ΔRep2 and Scr images, and "patch" pattern examples shown in the control images and the first ΔRep2 and Scr images. Data are pooled from two to three experiments; Kruskal–Wallis $P < 0.0001$; Mann–Whitney ****$P < 0.001$. Source data are available online for this figure.

To better understand the biophysical properties of the CSP repeats, we utilized optical tweezers combined with confocal single-molecule fluorescence microscopy (Hohng et al, 2007) and performed single-molecule fluorescence-force spectroscopy as previously performed for spider silk (Grashoff et al, 2010; Brenner et al, 2016). Because phenotyping of the CSP repeat mutants suggested that both the repetitive structure and length of the repeats were critical to their function, we tested different-length repeat peptides and a scrambled repeat peptide in these experiments. Single-molecule FRET (Ha et al, 1996) signal is measured between two fluorophores, Cy3 (FRET donor) and Cy5 (FRET acceptor), flanking the peptide of interest while the peptide is under tension. One end of the peptide-fluorophore construct is immobilized on a passivated glass coverslip, while the other end is conjugated to a bead in an optical trap through DNA linkers (Fig. 8A). Upon moving the microscope stage, force is applied to the peptide and peptide extension can be measured by changes in the FRET efficiency. As shown in Fig. 8A, when the peptide is

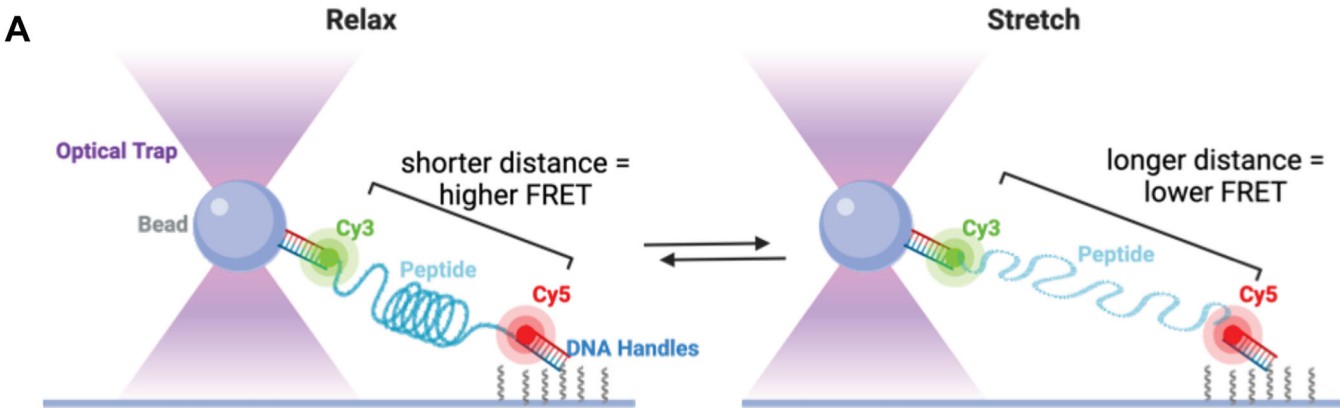

**Figure 8.  CSP repeats have elastic properties.**

Wild-type and scrambled CSP repeat peptides were synthesized and prepared for single-molecule force spectroscopy studies with the addition of DNA handles and fluorophores. (A) Schematic of single-molecular force spectroscopy (SMFS) setup showing one end of the peptide conjugated to a DNA handle that then attaches to a bead caught in an optical trap, and the other end of the peptide is immobilized on a glass slide. As the bead is pulled, it stretches the peptide, increasing the distance between the fluorophores, thus decreasing the FRET signal. Upon relaxation, the fluorophores are brought closer together, and the FRET signal increases. Peptide sequences used in study are shown below the SMFS setup. (B, C) FRET signal during stretch (black) and relax (red) phases of wild-type $CSP_{24}$ peptide pulling $n = 22$ and scrambled peptide pulling $n = 16$. Shown are means $+/-$ standard error. (D) A longer wild-type peptide $CSP_{40}$ repeat sequence was prepared and tested in single-molecule fluorescence force spectroscopy studies. FRET signal during stretch (black) and relax (red) phases of $CSP_{40}$ peptide pulling. Error bars are means $+/-$ standard error, $n = 22$. (E) Wild-type CSP repeats act as a linear spring independent of SMCC linker. Compliance (the distance the peptide stretched upon increasing force after linker subtraction) of the peptides is dependent on length and fits a linear fit model, a feature of linear springs. Two wild-type peptide lengths were tested: $CSP_{40}$ and $CSP_{24}$. Lines = liner fit model. $R^2$ values for $CSP_{24} = 0.93$ and for $CSP_{40} = 0.86$. Error bars are means $+/-$ standard error, $n = 22$ for both peptides. Source data are available online for this figure.

relaxed the FRET signal is high, as the Cy3 and Cy5 fluorophores are in proximity, while upon application of force, the fluorophores move apart as detected through reduction in FRET. FRET efficiency is restored to the high value when the peptide is relaxed to the original low force. With elastic peptides that can stretch without heat dissipation, FRET vs force curves of the stretching and relaxation phases should overlap, which is indeed what we observed for a 24 amino acid CSP repeat peptide, $(NANDPAPP)_3$, henceforth referred to as $CSP_{24}$ (Fig. 8B). In contrast, the scrambled peptide had a relaxation phase with a lower FRET efficiency than during the stretching phase, indicating that the peptide does not rapidly recover its original compact state during force relaxation (Fig. 8C). Such hysteresis indicates that the scrambled peptide behaves as an inelastic spring and that the elastic behavior observed for the CSP repeat peptide is a biophysical feature of its specific arrangement of amino acids. A similar elastic behavior without hysteresis was also observed for a longer 40 amino acid CSP repeat peptide $(NANDPAPP)_5$, referred to as $CSP_{40}$ (Fig. 8D), but the FRET efficiencies are lower across all force values compared to $CSP_{24}$ due to the increased peptide length. Finally, we converted FRET efficiency into the end-to-end distance of the peptide after correcting for the linkers between the fluorophores and the peptide, and found that the CSP repeats are linear springs, i.e., their extension increases linearly with force (Fig. 8E). Compliance, defined as the extension change per unit force change, is larger for the longer peptide, 0.08 nm/pN for $(NANDPAPP)_5$ vs 0.048 nm/pN for $(NANDPAPP)_3$, further confirming that our experimental scheme indeed measures the mechanical properties of the CSP repeat peptides. The peptide-length normalized compliance values, 0.02 nm/pN/amino acid, for both $(NANDPAPP)_5$ and for $(NAND-PAPP)_3$, are much larger than those measured from spider silk peptides, ~0.0012 nm/pN/amino acid (Brenner et al, 2016), suggesting that the CSP repeat peptides are much stiffer. Thus while like the spider silk protein, the CSP repeat peptides are elastic in that they can unfold and fold without hysteresis, a higher force is required to stretch the CSP repeat peptides compared to the spider silk peptide. Together, these data suggest that the CSP repeat region has properties of a linear-elastic spring and that the elasticity of the region is dependent on the repeated, primary amino acid sequence and the length of the repeats.

Previous molecular dynamics simulations of peptides derived from *P. vivax* CSP repeats estimated the elastic modulus of the 27 amino acid peptides to be about 4 cal/(mol Å²), corresponding to the peptide length normalized compliance of about 0.055 nm/pN/amino acid (Kucharska et al, 2022). This is more than 20-fold larger

than what we estimated experimentally from *P. berghei* CPS repeats, 0.002 nm/pN/amino acid. Although we cannot rule out species differences as the source discrepancy, here we suggest an alternative model. From our data shown in Fig. 8, we obtained the end-to-end distance of the peptides in the absence of force, $R$, by correcting for the cross-linker length and extrapolating the linear fits to $F = 0$ pN (Fig. 8E). $R$ was 1.97 nm for $CSP_{40}$, which is ~1.6 times that of $CSP_{24}$ ($R = 1.23$ nm). Thus, $R$ is directly proportional to the number of amino acids, suggesting that CSP behaves like a rod-like, folded structure with a defined zero-force equilibrium length. The rod is extremely compact ($R/L_c = 0.15$, where $L_c$ is the contour length $= 0.35$ nm times the number of amino acids). Therefore, each amino acid contributes ~0.04 nm to the length of the peptide. This may hint at the presence of eight-amino acid-long periodic structures, of length ~0.32 nm each, which act as independent nanosprings in series.

Another previous study using an atomic force spectroscopy (AFM) approach also suggested an elastic CSP repeat region (Patra et al, 2017), the constructs used in that study had I27 domains from titin flanking the CSP repeats to provide a molecular fingerprint in the stress-strain curves so that the peaks associated with the CSP repeats could be identified. However, they found that the CSP repeats unfolded before the I27 bands, meaning that contributions of the I27 domains to the CSP elasticity data cannot be confidently excluded (Patra et al, 2017). As our constructs have no such domains and the compliance of the peptides is dependent on their length, we can confidently extrapolate the mechanical properties of the CSP repeats. Interestingly, the CSP repeats are substantially stiffer (~4×) than the previously characterized spider silk proteins, making them an interesting potential molecular tension-sensing tool (Brenner et al, 2016).

## Discussion

In this study, we investigated the structural and functional properties of the CSP repeats by generating mutant parasites with altered repeats and using single-molecule mechanical measurements. These orthogonal approaches both demonstrate that the length and organization of the repeats into repetitive sequence blocks are essential to their functional properties, putting to rest the long-held belief that the CSP repeats are a non-functional linker and without conserved structural properties (Schofield et al, 1990). Here we demonstrate that the repeats do indeed link the N- and C-terminal domains and that this property is dependent on their

length. Furthermore, though CSP is GPI-anchored and thus, not a component of the adhesion sites that link to the motor, the CSP repeats enable efficient formation and turnover of adhesion sites on the sporozoite's surface. We hypothesize that the repeats form a network, similar to other elastomeric proteins, that allows for the cohesion of the secreted adhesion site on the parasite's plasma membrane.

The repeat mutants generated in the current study develop normally in the oocyst, indicating that the mutations we introduced do not lead to gross misfolding of CSP and allowing for downstream functional analysis in salivary gland sporozoites. Mutants with a ~25% deletion of the repeats, ΔRep1 and ΔRep3, did not have significant phenotypes in our assays, while the 50% truncation mutant ΔRep2 and the scrambled repeat mutant, Scr, had gliding motility defects that impacted both salivary gland and hepatocyte infection. While the Scr and ΔRep2 had similar motility defects (Figs. 4–7), the ΔRep2 mutant was significantly more attenuated in both salivary gland and hepatocyte infection. We think this is likely due to the additional deficit observed in ΔRep2 sporozoites of increased exposure of the C-terminal TSR domain. This could explain their decreased salivary gland numbers as TSR exposure is known to lead to sticking of sporozoites to other mosquito organs (Coppi et al, 2011) and could impact hepatocyte infection, possibly via inefficient cleavage of the N-terminus upon arrival in the liver. Though we assessed CSP cleavage in these mutants (Appendix Fig. S4), the assay is not sufficiently sensitive to pick up differences in cleavage efficiency. While the differences between ΔRep2 and the mutants with smaller repeat deletions, ΔRep1 and ΔRep3, could be attributed to the specific sequences that were deleted, i.e., the second repeat sequence may be more important than the 1st and 3rd repeat sequences, we believe that it is more likely due to the length of the deletion for a few reasons: The scrambled repeat mutant in which the entire repeat region was altered did not have as severe a phenotype as ΔRep2, suggesting that the sequence of the 2nd repeat is not uniquely responsible for the ΔRep2 phenotype. Furthermore, the sequence of the 1st repeat is very similar to the 2nd, suggesting the size of the deletion is important. Future experiments in which the entire repeat is engineered to have the sequence of the 2nd repeat and smaller truncations in this mutant would be informative as to whether this is definitively the case.

Here, we demonstrate that mutant sporozoites with truncated or scrambled repeats have a diminished capacity for productive motility, with an increased proportion of sporozoites patch gliding or waving. RICM imaging demonstrated severe adhesion site formation defects in patch-gliders, a gliding phenotype rarely observed in wild-type salivary gland sporozoites and commonly observed in ΔRep2 and Scr mutants. Additionally, the small percentage of circular gliding ΔRep2 and Scr sporozoites had defects in adhesion site formation and turnover. Our data further demonstrate that the defect in adhesion site formation is due to dysfunction on the sporozoite surface and not to problems with the internal motility machinery. Our finding that patch gliding ΔRep2 and Scr sporozoites move faster than control sporozoites and that circular gliding mutants sporozoites turn over their adhesion sites at a slower rate, yet reach maximum speeds comparable that of controls, all suggest that the subpellicular actin-myosin motor in these mutants is fully developed. Furthermore, the normal secretion of TRAP onto the parasite surface in ΔRep2 and Scr sporozoites,

suggests the gliding defects in these mutants are downstream of TRAP secretion.

TRAP, the dominant adhesin during sporozoite motility, must be shed from the surface to allow the parasite to disengage from adhesive interactions and move forward, a process that requires the activity of plasma membrane rhomboid proteases (Buguliskis et al, 2010; Ejigiri et al, 2012; Shen et al, 2014). Our finding of increased TRAP on the surface of ΔRep2 and Scr sporozoites is similar to what was observed in sporozoites in which the rhomboid cleavage site of TRAP was mutated (Ejigiri et al, 2012). However, in contrast to the CSP repeat mutants, the TRAP cleavage site mutants form adhesion sites from which they have trouble disengaging, leaving thick TRAP trails in their wake (Ejigiri et al, 2012). In distinction, the formation of mature adhesion sites in the CSP repeat mutants rarely occurs, as evidenced by the increased proportion of patch gliders and decreased adhesion site turnover in the small proportion of ΔRep2 and Scr circular gliders. The lack of TRAP trails in the CSP repeat mutants suggests that without a proper adhesion site, the rhomboid protease may not have access to TRAP. Thus, while the majority of rhomboid cleavage site mutant sporozoites engage in productive, albeit slower, motility, only a few of the ΔRep2 and Scr mutants are capable of productive motility, suggesting that what happens to TRAP after its secretion onto the sporozoite surface differs between these two sets of mutants.

Visualization of MTIP patches on the surface of ΔRep2 and Scr mutants further supports the finding that adhesion site formation is defective in these mutants. Unfortunately, it has been difficult to visualize TRAP during gliding motility: GFP-tagging of TRAP results in non-motile sporozoites (Kehrer et al, 2016), and fixation of gliding sporozoites does not reproducibly demonstrate TRAP in patches, likely due to its movement during the fixation procedure. However, we and others have demonstrated that components of the motor that interact with the inner membrane complex, such as MTIP and GAP40, are stable during fixation and can be visualized in patches on the gliding sporozoite's surface (Siden-Kiamos et al, 2020; Swearingen et al, 2016). MTIP staining on gliding control and mutant sporozoites showed that, in contrast to controls, MTIP was less frequently observed in patches and when present, the patches were significantly smaller on ΔRep2 and Scr sporozoites, supporting the hypothesis that proper adhesion sites are not formed on the surface of these mutants. Taken together, our data suggest that the motility defects observed in the ΔRep2 and Scr repeat mutants are due to abnormal adhesion site formation and dynamics, resulting from alterations in the organization of CSP on the sporozoite surface. We hypothesize that though not a component of adhesion sites, the CSP repeats form a higher order structure to provide an environment in which adhesion sites can form, with the components of the adhesion site assembling and remaining in association with one another within the dense surface coat of CSP (Fig. 9A). Interestingly, a previous study in which optical tweezers were used to probe the sporozoite surface, found spatially segregated functional effects on motility suggesting that there are distinct cohesive capacities on the sporozoite surface (Hegge et al, 2012).

Deletion of 25% of the repeat region, whether it be in the N-terminal (ΔRep1) or the C-terminal (ΔRep3) portion of the repeats, does not significantly impact function. In contrast, deletion of 50% of the repeat region (ΔRep2) leads to the motility defects

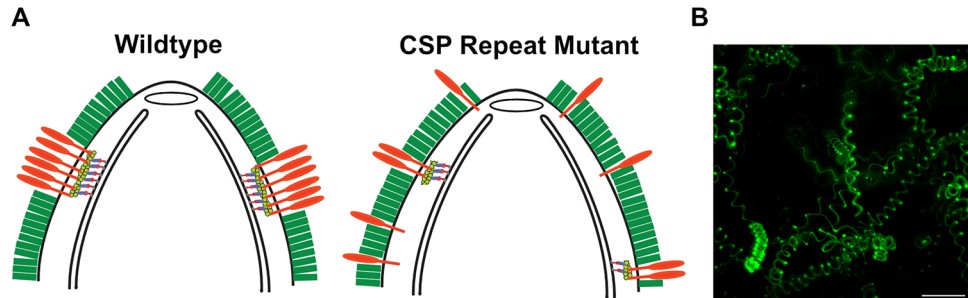

**Figure 9. Sporozoite deformability in 3-D matrix and model of adhesion site formation in wild-type and mutant sporozoites.**

(A) Cartoon of the anterior portion of a wild-type and CSP repeat mutant sporozoite, with CSP (green) on the plasma membrane and TRAP (orange), forming a large adhesion site in wild-type sporozoites, or being distributed as small adhesion sites throughout the mass of surface CSP. Underneath the plasma membrane, a simplified motor complex embedded in the inner membrane complex is shown. Smaller actin filaments (shown in light green, attached to the TRAP cytoplasmic domain) are likely to result from smaller adhesion sites, thus impacting motility. (B) Maximum projection of a 1-min movie showing tracks of wild-type *P. berghei* sporozoites as they move in 3-dimensions through matrigel. Note the different slopes of the helices. Scale bar = 50 µm. Source data are available online for this figure.

outlined above and increased TSR-domain exposure. Since the TSR domain is normally masked in salivary gland sporozoites (Coppi et al, 2011; Hopp et al, 2015), our data suggest that there is a length and/or conformational requirement for masking of the TSR. In field isolates, the number of CSP repeats can vary up to ~12% and modeling of repeat region length has suggested that length may be important for stabilizing repeat region structure (Escalante et al, 2002; Gandhi et al, 2014). The elastic properties of the CSP repeats could play a role in the masking of the TSR domain, with the 50% truncation in ΔRep2 not allowing for sufficient stretch to mask this domain.

Our biophysical studies demonstrating that the CSP repeats have elastic properties and behave as a stiff, linear spring are concordant with the motility defects of the repeat mutants. Both normal gliding motility and the biophysical properties of the repeats are dependent upon length and their organization into repetitive blocks. While we do not directly demonstrate a connection between the elastomeric properties of the repeats and their role in motility, we hypothesize that these properties are important in the phenotypes we observed in our mutants. Previous studies have found that proteins with elastomeric properties interact with one another to form higher-order structures that confer new mechanical and structural properties critically important to their function (Tatham et al, 2000). Thus, these properties may be important for both the organization of CSP on the sporozoite's surface, allowing for the formation and organization of proper adhesion sites within the dense CSP coat, and for the regulated exposure of the TSR domain. Studies exploring whether the elastomeric properties of the repeats orchestrate the separation of CSP from adhesion sites on the sporozoite surface would be of particular interest. It is also possible that the elastic properties of the repeats modulate the dramatic changes in sporozoite curvature as they migrate through three-dimensional matrices in vivo. Alternatively, they might enable the parasites to keep substrate contact as they bend upon interaction with the environment. 3D-projections of wild-type sporozoites migrating through matrigel (Fig. 9B) illustrate the curvature changes they experience during motility in vivo. Identifying the strength of the molecular forces experienced by the sporozoite, and its surface proteins, during

adhesion and migration could elucidate the functional importance of an elastic repeat region.

In conclusion, the finding that mutations to the CSP repeats cause organizational changes on the sporozoite's surface sufficient to impact adhesion site formation supports a long-held, but unproven, idea that CSP forms a higher order structure on the sporozoite surface (Ozaki et al, 1983). Our current model is that CSP, by forming a cohesive entity, allows the components of adhesion sites to assemble in proximity to one another rather than be randomly distributed in a "sea" of CSP. This model is supported by the dense packing of CSP on the sporozoite surface and the observation that CSP is shed as an intact coat upon cross-linking with antibody (Vanderberg et al, 1969).

## Methods

**Reagents and tools table**

| Reagent/resource | Reference or source | Identifier or catalog number |
|---|---|---|
| **Experimental models** | | |
| *Plasmodium berghei* ANKA C57cl1 | PMID: 16242190 | |
| *Anopheles stephensi* mosquitoes, Liston strain | BEI Resources | |
| Hepa1-6 cells | ATCC | CRL-1830 |
| **Recombinant DNA** | | |
| **Antibodies** | | |
| mAb 3D11 | PMID: 6985745 | |
| Anti-C-terminal TSR domain of CSP | PMID: 21262960 and PMID: 15630135 | |
| Anti-TRAP repeat region | PMID: 22911675 | |
| Anti-CSP C-Dis | This publication | |
| **Oligonucleotides and other sequence-based reagents** | | |
| *csp* repeat region oligonucleotides | Genscript, Piscataway, NJ | |

| Reagent/resource | Reference or source | Identifier or catalog number |
|---|---|---|
| **Chemicals, enzymes, and other reagents** | | |
| [35S]Cys/Met | MP Biomedicals | 0151006 |
| Matrigel | Becton Dickenson | 356231 |
| **Software** | | |
| Image J | | |
| NIS Elements | Nikon | |
| **Other** | | |

## Ethics statement

All animal work was conducted in accordance with the recommendations in the Guide for the Care and Use of Laboratory Animals from the National Institutes of Health. This work was approved by the Johns Hopkins University Animal Care and Use Committee (protocols RA110608, M011H467 and M014H363), which is fully accredited by the Association for the Assessment and Accreditation of Laboratory Animal Care.

## Antibodies

Anti-C-terminal sera were previously generated against a 100 amino acid peptide representing the C-terminal third of CSP (Coppi et al, 2005). mAb 3D11 is specific for the *P. berghei* CSP repeats (Yoshida et al, 1980), and mAb 2E6 is specific for *Plasmodium* Hsp70 (Tsuji et al, 1994). Antisera against the *P. berghei* TRAP repeats are specific for the C-terminal repeat region of TRAP (Ejigiri et al, 2012). Antisera to a predicted disordered region immediately C-terminal of the CSP repeat region and upstream of the TSR (anti-CSP C-Dis), was generated in a New Zealand White rabbit using the peptide CDDSYIPSAEKILEFVKQIRDSITE, synthesized and purified by RS Synthesis (Louisville, KY). The peptide was conjugated to keyhole limpet hemacyanin, and the rabbit was immunized once and boosted three times at ≥1 month intervals as previously outlined (Harlow et al, 1988).

## Plasmid generation

Mutant *csp* repeat region oligonucleotides were commercially synthesized by Genscript (Piscataway, NJ) and provided in the pUC57 plasmid. Repeat sequences were digested from the pUC57 plasmid using HindIII-HF and SexAI and ligated into an intermediate CSP vector in Bluescript SK, containing the KpnI to PacI sequence from the *csp* locus, which includes 500 bp of 5′utr and the full-length *csp* sequence. The *csp* sequence in this vector was engineered to have HindIII-HF and SexAI restriction sites flanking the repeats. Once the mutant repeat sequences were ligated into the intermediate CSP vector, a KpnI and PacI digest was performed and this fragment was ligated into the previously described transfection vector, pCSRep, altered to have an additional 1 kb of 3' UTR in order to minimize correction of the repeat mutations (Coppi et al, 2011; Ferguson et al, 2014) (Appendix Fig. S1A).

## Generation and verification of *P. berghei* CSP repeat mutants

Recombinant parasites were generated in the *P. berghei* ANKA 507 cl1 line, which expresses GFP under the *eef1α* promoter at the 230p locus (WT-GFP) (Janse et al, 2006b). Recombinant parasites were generated by double homologous recombination in which the native *csp* locus was replaced by a mutant copy of *csp* with its control elements and an upstream selection cassette (Appendix Fig. S1). pCSRep was digested with XhoI and KasI and 5–10 μg of DNA was electroporated into schizonts which were injected intravenously into a Swiss Webster mouse, selected with pyrimethamine, and cloned by limiting dilution in Swiss Webster mice as previously described (Janse et al, 2006a). Repeat mutant genotypes were confirmed by PCR and sequencing (Appendix Fig. S1). 5′ integration was confirmed using primers DP1 (5′-AATGAGACTATCCCTAAGGG-3′) and DP2 (5′-TAATTATATGTTATTTTATTTCCAC-3′), 3′ integration was confirmed using primers P6 (5′-TGATTCATAAAT-AGTTGGACTT-GATTT-3′) and P9 (5′-TCGAAATGGGCGCTGAC AAGAA-3′) and amplification of the *csp* locus was used to confirm repeat mutations using primers P3 (5′-CCATTTTAGTTGTAGCGTCACTTTT-3′) and P4 (5′-ACAAATCCTAATGAATTGCTTACA-3′). Absence of WT parasites in mutant clones was confirmed using size shift PCR. P14 (5′-CGTGCATTTTGTGTCCTCATGTTGC-3′), binds in the 5′ UTR of *csp*, and P17 (5′-GCTCGTTTAAGTTCCTTTGGGCTTGG-3′), binds the 5′ end of *csp*. This primer pair resulted in a 1.46 kb product in WT parasites and a 4.8 kb product in recombinant parasites. After PCR confirmation, *csp* was amplified by PCR and sent for sequencing.

## Mosquito infection and sporozoite isolation

*Anopheles stephensi* mosquitoes were reared in the insectary at the Johns Hopkins Bloomberg School of Public Health and infected with control and mutant parasite lines by feeding on anesthetized Swiss Webster mice as previously outlined (Coppi et al, 2011). Salivary gland sporozoites were harvested from salivary glands dissected in Lebowitz media (L-15, Gibco #11415-064) 21–25 days after mosquito feeds. Dissected salivary glands were briefly spun to remove the media and add fresh L-15 prior to releasing sporozoites by homogenization. Sporozoite preps were then centrifuged at 4 °C for 4 min at $100 \times g$, and sporozoites were collected from the supernatant.

## Quantification of midgut and salivary gland infections

Oocyst numbers were assessed at 10–14 days post blood meal. Mosquito midguts were dissected, mounted on a glass slide and imaged with a ×4 objective on a Nikon Eclipse E600 upright microscope and counted using the GFP fluorescence of parasites. Quantification of sporozoites in the midgut, hemolymph, and salivary glands was performed on mosquitoes asorted by GFP fluorescence. Twenty midguts and salivary glands were dissected on day 16 or 21 post blood meal, respectively, homogenized to release sporozoites, which were then counted on a hemocytometer. Hemolymph was collected by perfusion of the mosquito abdomen on day 18 post blood meal, and sporozoites were counted as above. For all sporozoite counts, the total number of collected sporozoites

was divided by the number of mosquitoes to calculate the average number of sporozoites/mosquito. Salivary glands of ΔRep2 and Scr-infected mosquitoes were imaged to confirm that sporozoites invaded the glands. Salivary glands were cleanly dissected from infected mosquitoes placed into 5 μl of L-15 on a MaTek well (MaTek #P35G-0-14C) followed by the addition of 5 μl of matrigel (BD #356231). The matrigel was allowed to polymerize at room temperature for ~10 min and Z-stacks of salivary glands were acquired using a 3i spinning disk confocal microscope (Zeiss AxioObserver Z1 microscope with Yokogawa CSU222 spinning disk), with a 472 nm laser to observe GFP-expressing sporozoites and DIC to observe the gland architecture.

## Western blots

Anti-CSP C-Dis, specific for the sequence just upstream of the *P. berghei* TSR domain, DDSYIPSAEKILEFVKQIRDSITE, was generated as outlined in the "Antibodies" section above. This sequence represents a predicted disordered region between the repeats and TSR of *P. berghei* CSP was used for CSP quantification by western blot since antibodies specific for the repeat region could not be used in our mutants. For ΔRep1, ΔRep3, and RCon lines, salivary gland sporozoites were isolated as described, and for ΔRep2 and Scr lines, hemolymph sporozoites were used, as low parasite abundance in the salivary glands prevented clean western blots. Sporozoites were pelleted, resuspended in sample buffer (0.125 M Tris-HCl, 20% glycerol, 4% SDS, 0.002% bromophenol blue, 50 mM DTT, and 1× protease inhibitors (Roche #11-836-153-001) at a concentration of $10^4$ sporozoites/μl, and 100,000 sporozoites were loaded per well. Samples were run on a 10% SDS-PAGE gel and transferred to a nitrocellulose membrane. CSP was detected using anti-CSP C-Dis polyclonal sera (1:100) and TRAP, used as a loading control, was detected with anti-TRAP polyclonal sera (1:500) (Ejigiri et al, 2012). Western blots were developed with goat anti-rabbit HRP antibody (1:10,000; GE Healthcare #NA934V) and ECL reagent (GE Healthcare #RPN2106).

## Immunofluorescence assays (IFAs)

Salivary gland sporozoites were isolated from mosquitoes as described above and spun onto 12-mm coverslips in a 24-well plate at $300 \times g$ for 3 min at 4 °C with low acceleration and no brake. Sporozoites were fixed for 1 h with 4% PFA at room temperature, washed, blocked with 1% BSA/PBS, and incubated at 37 °C with anti-C-terminal sera (1:100), specific for the *P. berghei* TSR domain (Coppi et al, 2011) in 1% BSA/PBS. When indicated, 0.1% saponin was incorporated into primary antibody dilutions to break intramolecular interactions on the surface of the sporozoite. For TRAP staining, a "gliding IFA" was performed. After sporozoites were spun onto coverslips, media was exchanged for 2% BSA/L-15, pH 7.4 and incubated for 1 h at 37 °C with 5% CO$_2$. After this, they were fixed as above, blocked in 1% BSA/5% goat serum/PBS and incubated with anti-TRAP repeat antisera (1:100) in 1% BSA/5% goat serum/PBS. After incubation with the appropriate secondary antibody, coverslips were mounted with Prolong Gold, and image acquisition was performed using a 100x objective on a Nikon Eclipse E600 upright microscope with a DS-Ri1 digital camera under identical acquisition settings for each treatment condition. Sporozoites were identified by phase-contrast,

and then fluorescence imaging was performed to avoid biased acquisition. Intensity measurements were quantified using NIS Elements Br 3.2 software by using the "Auto ROI" function to identify the sporozoites in phase before switching to the fluorescence channel for intensity quantification. For MTIP and TRAP staining, freshly dissected sporozoites were spun onto 12-mm coverslips in a 24-well plate at $300 \times g$ for 5 min and allowed to glide at 37 °C for 30 min in 1% BSA in L-15, pH 7.4. Sporozoites were then fixed for 1 h with 4% PFA at room temperature, blocked with 1% BSA/PBS for 30 min at room temperature and incubated with rabbit polyclonal anti-MTIP (1:500) (Bergman et al, 2003) or anti-TRAP (1:100) in 1% BSA/PBS followed by detection with Alexa fluor 546 goat anti-rabbit secondary antibody (1:500, Invitrogen #A11010) in PBS. Samples were mounted in gold antifade mountant (Invitrogen, P36935), and image acquisition was performed using a Nikon Eclipse E600 (for TRAP staining) or a confocal Zeiss LSM 800 with a 488 and 561 laser (for MTIP staining).

## Metabolic labeling

Sporozoites were metabolically labeled and CSP was immunoprecipitated and detected as previously outlined (Coppi et al, 2005). Briefly, mosquito dissections were performed using DMEM without Cys/Met (Corning #17-204-C1) and sporozoites were metabolically labeled in DMEM without Cys/Met, 1% BSA and 400 μCi/ml l-[35S]Cys/Met (MP Biomedicals #0151006) for 45 min (ΔRep1) or 1.5 h (ΔRep2) at 28 °C. Labeled sporozoites were washed three times, lysed, and CSP was immunoprecipitated with mAb 3D11 conjugated to agarose beads, eluted from beads with 0.1 M glycine, pH 1.5 and 1% SDS and analyzed by SDS-PAGE followed by autoradiography.

## In vivo sporozoite infectivity assays

In all, 5000 sporozoites were injected intravenously into mice in 200 μL of cell culture media. For pre-patency experiments, blood smears were made from days 3–9 after inoculation of sporozoites, stained with Giemsa (Sigma-Aldrich #GS500), and screened for 5 min for the presence of parasites. For liver load experiments, mice were sacrificed 39 h post sporozoite inoculation, livers were harvested, and RNA was extracted as previously outlined (Bruna-Romero et al, 2001). Parasite liver load was quantified by RT-qPCR using primers specific for *P. berghei* 18s rRNA and compared to a plasmid standard curve. In all, 4–5-week-old Swiss Webster mice (Taconic) were used for pre-patency experiments and 4–5-week-old C57BL/6 mice (Taconic) were used for liver load experiments.

## In vitro hepatocyte infection assays

Hepa1-6 cells (ATCC #CRL-1830), a mouse hepatocyte cell line, were seeded on collagen I-coated LabTek wells (LabTek #17745) with 100,000 cells/well in 400 μl of cell culture media (DMEM, 10% fetal calf serum, and 2 mM gentamycin) 1 day before infection with sporozoites. In total, 20,000 sporozoites were added to each well and centrifuged onto cells at $300 \times g$ for 3 min at room temperature with low acceleration and no brake. Sporozoites were then allowed to invade hepatocytes for 1 h at 37 °C with 5% CO$_2$, after which the media was replaced and cultures were maintained with 1×

penicillin-streptomycin (Gibco #15140-122). Media was changed twice daily, and at 24- or 48-h post-infection, hepatocytes were fixed with 4% PFA for 1 h at room temperature. Cells were permeabilized with 100% methanol for 30 min at −20 °C, followed by blocking with 1% BSA/PBS at 37 °C for 1 h. Parasites were stained with 10 µg/mL mAb 2E6 in 1% BSA/PBS for 1 h at 37 °C, followed by 30 min with 0.1% Tween/PBS at 37 °C and 1 h at 37 °C with secondary antibody Alexa-488 goat anti-mouse (1:500) (Life Technologies #A11029) in 1% BSA/PBS before mounting with Prolong Gold with DAPI (Invitrogen #P36935). The number of EEFs per 50 fields at 24 h post-infection was quantified for each well. For quantification of EEF size, the 2D area was measured using NIS Elements "Auto ROI" function on a Nikon Eclipse E600 microscope with a DS-Ri1 digital camera.

## In vitro live gliding assays

Salivary gland sporozoites were dissected, and a concentrated sporozoite suspension was mixed with an equal volume of 2% BSA (Sigma #A7888) in L-15, pH 7.4 and incubated at 37 °C for 5 min to activate sporozoites. Activated sporozoite suspension was placed between a glass slide or 22 mm × 50 mm cover glass and a 22 mm × 50 mm cover glass. Sporozoites were allowed to settle for 2 min before imaging. These assays were performed using a Nikon Eclipse E600 upright microscope using a ×40 phase objective and imaged using phase contrast microscopy. Sporozoites were manually tracked using Fiji (https://fiji.sc). Sporozoite motility behavior was categorized as patch gliding if parasites moved over a single point without net displacement, as waving if sporozoites were attached without movement over the attachment site, and as circular gliders if they moved in circles, with displacement from the point of origin. Almost all sporozoites exhibited one form of motility during image acquisition. On the rare instances where sporozoites switched modalities, they were classified as the modality that is the most migratory (i.e., if a sporozoite did waving and circular gliding, it was classified as circular gliding). Sporozoite speed between each frame, instantaneous speed, was outputted using the Fiji "Manual Tracking" plugin. Frequency distributions were made by combining the instantaneous speed data points from all sporozoites within a group of interest (i.e., circular gliders). A frequency distribution was generated using 0.5 µm/s speed bins and then plotted as a line graph for visualization.

For the Matrigel assay (Fig. 9), freshly isolated mCherry-expressing *P. berghei* sporozoites (Hopp et al, 2015) were resuspended in 50 µl of 2% bovine serum albumin BSA in HBSS pH 7.4. The sporozoite suspension was then mixed with Matrigel® matrix basement membrane (Corning 356231) at a 1:1 ratio, and the mixture was transferred to a glass-bottom 96-well plate (Greiner 655892). The plate was placed in a DeltaVision® imaging system enclosed within a microcell incubator set to 37 °C. After 15 min of incubation, sporozoite movement in three dimensions was imaged using a 20X objective. Images were acquired in three Z-stacks at 10 µm increments every 1.45 s for 300 cycles (7.25 min).

## Reflection interference contrast microscopy (RICM)

RICM was performed using a LSM 780 inverted Zeiss AxioObserver with 780-Quasar confocal module and high-sensitivity gallium arsenide phosphide detectors (GaAsP) with a ×63 (NA 1.4) oil objective and excitation with a 488-argon laser. A dichroic beam splitter (transmitted 80%, reflected 20%) was used to collect light emitted between 414 and 690 nm. All movies were acquired at 1 frame per second for 2.5 min. All movies were acquired at 1 frame per second for 2.5 min. For RICM analysis of adhesion sites, each frame was manually categorized as a sporozoite with a single or multiple adhesion sites. The percent of time with multiple adhesion sites was calculated by dividing the number of frames with multiple adhesion sites by the total number of frames (150) and multiplying by 100. Adhesion site turnover was determined by quantifying the number of adhesion sites per movie (2.5 min) that appeared at the anterior end and translocated to the posterior end of the sporozoite, followed by complete disassembly.

## Single-molecule fluorescence-force spectroscopy

To generate the peptide-oligomer constructs with Cy3 and Cy5 at either end, the protocol was adapted from a previous study (Brenner et al, 2016). Briefly, the DNA-oligonucleotide, ACCGCTGCCGTCGCTCCG with a 5′ amine modification, was incubated with 200× molar excess of succinimidyl 4-[N-maleimi-domethyl]cyclo-hexane-1-carboxylate (SMCC-Sigma-Aldrich #M5525) for 2.5 h at room temperature to generate maleimide-DNA capable of sulfhydryl-reactive cross-linker chemistry with peptides with terminal cysteines. SMCC-Oligo conjugates were isolated using ethanol precipitation and dissolved in T150 buffer (25 mM Tris, 150 mM NaCl, 1 mM EDTA), and then run through Bio-Spin 6 columns (Bio-Rad #7326002) twice. Peptides with terminal cysteines were synthesized by GenScript and diluted to 200 µM in T150 buffer: WT (C-NANDPAPPNANDPAPPNAND-PAPP-C) or Scrambled (C-NNNAPDDNAAAANPPANPPPPPD-C). The peptide was added to SMCC-Oligo at a 1:2.5 molar ratio and incubated overnight at 4 °C for conjugation. Unreacted SMCC-Oligo was removed by polyacrylamide gel-electrophoresis followed by extraction from the gel. 250 pmoles of oligo-peptide conjugate was added to 250 pmoles 5′-biotin-TGGCGACGGCAGCGAGGC-Cy5-3′ and 300 pmoles 5′-GGGCGGCGACCTGCTGGGTAGTC-Cy3-3′ and incubated overnight at 4 °C for conjugation by complementary base pairing with the DNA conjugated to the peptide. For fluorescence force experiments, 16 nM λ-DNA (NEB #N3011S) in 0.120 M NaCl was heated to 80 °C for 10 min followed by 5 min on ice. The Cy3-Cy5-peptide construct was added at 10 nM with 0.2 mg/mL BSA and incubated at 4 °C for 3 h followed by addition of 200 nM 5′-AGGTCGCCGCCCTTT- digoxygenin-3′ and overnight incubation at 4 °C.

For single-molecule fluorescence-force spectroscopy experiments, an imaging chamber was made by sandwiching polyethylene-glycol (PEG) (a mixture of PEG-valeric acid and biotin-PEG-valeric acid, Laysan Bio) between a passivated quartz slide and coverslip. The imaging chamber was then incubated with 0.2 mg/mL neutravidin (Pierce) in blocking buffer (10 mM Tris-HCl pH 7.5, 50 mM NaCl, 1 mg/mL BSA, 1 mg/mL tRNA (Ambion)) for 1 h. The peptide constructs were then diluted to 20 pM and immobilized on the surface via biotin-neutravidin interaction. 1 M anti-digoxygenin-coated polystyrene beads (Poly-sciences) were conjugated to the peptide construct through λ-DNA of the peptide construct by incubation in a buffer containing 10 mM Tris HCl pH 7.5 and 150 mM NaCl for 30 min. Microscope setup consisted of a trapping laser (1064 nm, 800 mW, Spectra

Physics) to catch a bead. As the microscope stage was moved along the X- plane, a confocal laser (532 nm, 30 mW at maximum average power (World StarTech)) was used to follow Cy3/Cy5 emission profiles. To apply tension, the stage was moved between 14 and 16.8 μm at a constant rate of 445 nm/s as Cy3/Cy5 emission profiles were measured with an exposure time of 20 milliseconds. The applied force was measured by a quadrant photodiode (UDT/SPOT/9DMI) from the position of the tethered bead. FRET trajectories as a function of force applied to peptides were then binned by 0.5 pN increments and plotted using Origin software (OriginLab).

## Statistical analysis

Statistical analysis were performed in GraphPad Prism. All data were evaluated for skewedness using GraphPad Prism's statistical analysis. Datasets with skewedness values $<-1$ or $>1$ were considered non-parametric. For comparison between two groups either Mann–Whitney tests (non-parametric data) or Welch's $t$ tests (parametric data) were performed. For comparisons between multiple groups, a one-way ANOVA followed by Dunnett's post hoc test (parametric data) or Kuskal–Wallis followed by Mann–Whitney–Wilcoxon (non-parametric data) was used. For proportion data, a Z-score for population proportions was performed using the equation $\frac{(p_1 - p_2)}{\sqrt{\frac{p_2(1-p_1)}{n}}}$ with significance at $P < 0.01$. The tests used and $P$ values are described in the figure legends.

# Data availability

This study includes no data deposited in external repositories.

The source data of this paper are collected in the following database record: biostudies:S-SCDT-10_1038-S44318-025-00551-9.

# Peer review information

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

## Acknowledgements

We are grateful to the Insectary and Parasitology Core Facilities at the Johns Hopkins Malaria Research Institute, expertly staffed by Chris Kizito, Godfree Mlambo and Abhai Tripathi, and to the Bloomberg Family Foundation for their support of this facility. We are grateful to Barbara Smith and Dr. Scott Kuo of the Johns Hopkins University School of Medicine Microscope Facility for their invaluable assistance with the RICM imaging experiments. This study was supported by a fellowship from the Johns Hopkins Malaria Research Institute (AB), the National Institutes of Health grants R01AI056840 (PS) and R35 122569 (TH), and the Bloomberg Family Philanthropies. The RICM imaging reported in this publication was supported by the Office of the Director of the National Institutes of Health under award number S10OD016374.

## Author contributions

**Amanda E Balaban**: Conceptualization; Data curation; Formal analysis; Validation; Investigation; Visualization; Methodology; Writing—original draft. **Sachie Kanatani**: Data curation; Formal analysis; Validation; Investigation; Visualization; Methodology; Writing—review and editing. **Jaba Mitra**: Data curation; Formal analysis; Validation; Investigation; Methodology; Writing—review and editing. **Jason Gregory**: Investigation. **Natasha Vartak**: Investigation; Methodology. **Ariadne Sinnis-Bourozikas**: Data curation; Formal analysis; Investigation; Methodology. **Fredrich Frischknecht**: Methodology; Writing—review and editing. **Taekjip Ha**: Conceptualization; Resources; Data curation; Formal analysis; Supervision; Funding acquisition; Validation; Methodology; Writing—original draft; Project administration; Writing—review and editing. **Photini Sinnis**: Conceptualization; Resources; Data curation; Formal analysis; Supervision; Funding acquisition; Visualization; Methodology; Writing—original draft; Project administration; Writing—review and editing.

Source data underlying figure panels in this paper may have individual authorship assigned. Where available, figure panel/source data authorship is listed in the following database record: biostudies:S-SCDT-10_1038-S44318-025-00551-9.

## Disclosure and competing interests statement

The authors declare no competing interests.

