## [Peer Review File · The EMBO Journal]

The *Plasmodium* CSP repeats have elastic properties with an important role in sporozoite motility

Amanda Balaban, Sachie Kanatani, Jaba Mitra, Jason Gregory, Natasha Vartak, Ariadne Sinnis-Bourozikas, Friedrich Frischknecht, Taekjip Ha, and Photini Sinnis

Corresponding author(s): Photini Sinnis (psinnis1@jhu.edu)

Review Timeline:

Submission Date:	3rd Jan 25
Editorial Decision:	18th Feb 25
Revision Received:	7th May 25
Editorial Decision:	12th Jun 25
Revision Received:	22nd Jun 25
Accepted:	5th Aug 25

Editor: Ioannis Papaioannou

Transaction Report:

Dear Dr. Sinnis,

Thank you for submitting your manuscript EMBOJ-2025-120089 for consideration by The EMBO Journal, and for your patience during peer review. Your manuscript has now been seen by three experts in the field, and we have received the full set of their well-informed and detailed reports, which are included below. Please note that referee #2 has also sent us a copy of the manuscript with edits and suggestions, which is also attached to this e-mail message.

As you will see, the feedback of all three referees on your study is very supportive. They all find the manuscript novel and interesting, and addressing a relevant question in the field. They also point out that the experimental work is of high quality and the conclusions sufficiently supported by the presented data. The referees also identify a number of limitations and provide helpful and reasonable suggestions for the improvement of the study and the manuscript, which would further improve the clarity of the manuscript and increase the impact of the work on the field.

Given the referees' positive comments and recommendations, I would like to invite you to submit a revised version of the manuscript along with a detailed point-by-point response addressing all referees' comments. I should add that it is The EMBO Journal policy to allow only a single round of major revision, and acceptance of your manuscript will therefore depend on the completeness of your responses in this revised version. Please let me know if you have any questions or comments that you would like to discuss with me.

We generally allow three months as standard revision time (May 17, 2025). As a matter of policy, competing manuscripts published during this period will not negatively impact our assessment of the conceptual advance presented by your study. However, we request that you contact us as soon as possible upon publication of any related work, to discuss how to proceed. Should you foresee a problem in meeting this three-month deadline, please let us know in advance and we may be able to grant an extension.

Thank you for the opportunity to consider your work for publication in The EMBO Journal. I look forward to your revision.

Best regards,

Ioannis

Instructions for preparing your revised manuscript

1. When you are ready to submit the revision, please upload:

- A Word file of the manuscript text (including legends of main Figures, EV Figures and Tables). Please make sure that changes are highlighted (or "tracked") to be clearly visible.

- Individual production-quality figure files (one file per figure). When assembling your figures, please refer to our figure preparation guidelines in order to ensure proper formatting and readability in print as well as on screen:

If the data shown in a figure are obtained from n {less than or equal to} 2, please use scatter plots showing the individual data points.

i. the name of the statistical test used to generate error bars and P values

ii. the number (n) of independent experiments (please specify technical or biological replicates) underlying each data point

(discussion of statistical methodology can be reported in the Materials and Methods section, but figure legends should contain a basic description of n , P , and the test applied)

iii. the nature of the bars and error bars (s.d., s.e.m.).

- A point-by-point response to the referees' comments, with a detailed description of the changes made (as a word file). All

referees' concerns must be fully addressed and their suggestions taken on board. When preparing your letter of response to the referees' comments, please bear in mind that this will form part of the Review Process File and will therefore be available online to the community. Please note that you have the possibility to opt out of the transparent process at any stage prior to publication by letting the editorial office know (contact@embjournal.org); if you do opt out, the Review Process File link will point to the following statement: "No Review Process File is available with this article, as the authors have chosen not to make the review process public in this case.". For more details on our Transparent Editorial Process, please visit our website: <https://www.embopress.org/page/journal/14602075/authorguide#transparentprocess>

- Expanded View (EV) files (replacing Supplementary Information) that are collapsible/expandable online. A maximum of 5 EV Figures can be typeset. EV Figures should be cited as "Figure EV1, Figure EV2" etc. in the text, and their respective legends should be included in the manuscript file after the legends of regular figures. See detailed instructions regarding Expanded View files here:

- For the figures that you do NOT wish to display as Expanded View figures, they should be bundled together with their legends in a single PDF file called "Appendix", which should start with a short Table of Contents (including page numbers). Appendix figures should be referred to in the main text as: "Appendix Figure S1, Appendix Figure S2" etc. Please see detailed instructions here: <https://www.embopress.org/page/journal/14602075/authorguide#expandedview>

- A complete author checklist, which you can download from our author guidelines (<https://www.embopress.org/page/journal/14602075/authorguide>). Please note that the checklist will also be part of the Review Process File.

2. Please note that no statistics should be calculated and shown in Figures if $n=2$. Please also note that each p value should be reported as an exact value.

3. Before submitting your revision, primary datasets (and computer code, where appropriate) produced in this study need to be deposited in appropriate public databases (see <https://www.embopress.org/page/journal/14602075/authorguide#dataavailability>). The accession numbers, database, and the specific URLs (links) should be listed in a formal "Data availability" section (placed after Methods), following the example below:

"The RNA-seq datasets produced in this study are available in the following database:
Gene Expression Omnibus GSE46843 (<https://www.ncbi.nlm.nih.gov/geo/query/acc.cgi?acc=GSE46843>)"

*** All links should resolve to a page where the data can be accessed. ***

*** Please remember to provide in the Data availability section of your revised manuscript reviewer passwords if the datasets are not yet public. ***

*** The Data Availability Section is restricted to new primary data that are part of this study. In case you have no data that require deposition in a public database, please state so instead of referring to the database: "Our study includes no data deposited in public repositories." under the heading "Data availability". ***

4. Please check that the title and the abstract of the manuscript are brief, yet explicit, even to non-specialists. The length of the title should not exceed 100 characters, and the abstract should be a single paragraph not exceeding 175 words.

5. Please also note our reference format: <https://www.embopress.org/page/journal/14602075/authorguide#referencesformat>.

7. Please remember: digital image enhancement is acceptable practice, as long as it accurately represents the original data and conforms to community standards. If a figure has been subjected to significant electronic manipulation, this must be noted in the figure legend or in the "Materials and Methods" section. The editors reserve the right to request original versions of figures and the original images that were used to assemble the figure.

8. Our journal encourages inclusion of data citations in the reference list to directly cite datasets that were obtained from public databases. Data citations in the article text are distinct from normal bibliographical citations and should directly link to the database records from which the data can be accessed. In the main text, data citations are formatted as follows: "Data ref: Smith et al, 2001" or "Data ref: NCBI Sequence Read Archive PRJNA342805, 2017". In the Reference list, data citations must be labeled with "[DATASET]". A data reference must provide the database name, accession number/identifiers, and a resolvable link to the landing page from which the data can be accessed at the end of the reference. Further instructions are available at: <https://www.embopress.org/page/journal/14602075/authorguide#referencesformat>.

9. We request authors to consider both actual and perceived competing interests. Please review our policy (<https://www.embopress.org/page/journal/14602075/authorguide#conflictofinterest>) and update your competing interests statement if necessary. Please name this section 'Disclosure and competing interests statement' and place it after the Acknowledgements section.

10. Please note that all corresponding authors are required to provide an ORCID ID upon submission of a revised manuscript (<https://orcid.org/>). Please find instructions on how to link your ORCID ID to your account in our manuscript tracking system in our Author guidelines (<https://www.embopress.org/page/journal/14602075/authorguide#authorshipguidelines>).

11. We use CRediT to specify the contributions of each author in the journal submission system. CRediT replaces the author contribution section, which should be removed from the manuscript. Please use the free text box to provide more detailed descriptions. See also guide to authors: <https://www.embopress.org/page/journal/14602075/authorguide#authorshipguidelines>.

13. We would also welcome the submission of cover suggestions or motifs to be used by our Graphics Illustrator in designing a cover.

14. Please use the link below to submit your revision:
<https://emboj.msubmit.net/cgi-bin/main.plex>

Referee #1:

ARTICLE SUMMARY

In their work, Balaban et al. present an investigation of the molecular properties of the circumsporozoite protein (CSP), the major surface antigen of the malaria sporozoite that plays a central role in its (immuno)biology. The work's focus mainly lies in characterizing the role of the CSP repeat region, a relatively long stretch of tandem repeats of which the exact composition and length varies with the Plasmodium species. The study's premise is that the organization of the CSP repeat region supports a higher order architecture of the antigen at the parasite surface, which is proposed to be important for sporozoite biology. The authors have employed *P. berghei* CSP as a model system and provide mainly functional evidence, supported by biophysics. The functional assays consist in generating sporozoites with wild-type CSP, truncated CSP (Δ Rep variants), or scrambled CSP, followed by an assessment of their phenotypes with regards to salivary gland invasion, antibody responses, motility and infectivity. The biophysical support is provided by single-molecule force spectroscopy, which yields insights into the elastic properties of the CSP repeat region. Finally, the findings are discussed within the context of the existing CSP literature.

DETAILED REPORT - OVERALL CONSIDERATION

The paper is well written and the results are presented in a logical fashion. The experiments seem technically well executed and the data appear to be carefully analyzed. The figures are clear and the provided movies are an absolute added value to the manuscript. The presented findings are novel and relevant. While most manuscripts in the literature have focused on specific CSP domains or regions, studies on the overall architecture of the full-length protein are scarce (despite decades of research). The paper provides valuable insights into the functional role of CSP's molecular organization under native conditions at the parasite surface. The choice for *P. berghei* CSP is logical as this represents a tractable model. The findings obtained for *P. berghei* CSP are of additional interest as some of these molecular principles could apply to the functional architectures of CSPs from human malaria parasites (e.g., *P. falciparum* and *P. vivax*). My suggestions to adapt the manuscript are related to changes in the text and to dig deeper into the existing data sets. This reviewer sees no need for additional wet lab experiments.

I will outline my comments as detailed as possible to give the authors the possibility to take these suggestions on board. My comments/suggestions are presented in a constructive spirit, and I hope the authors will read them bearing this in mind.

COMMENTS RELATED TO THE INTRODUCTION

Comment 1: The introduction is well written. However, it is this reviewer's opinion that it lacks information on the structural features of CSP domains that have been thoroughly characterized in the past. As this journal has a relatively broad readership (including scientists that are not active within the malaria/CSP field), I think this could be of interest to the reader and therefore propose to provide more information on this topic. For example, the CSP C-terminal and N-terminal domains have been thoroughly structurally characterized by Doud et al. (PMID 22547819) and Geens et al. (PMID 38059674), respectively.

Comment 2: I kindly ask the authors to double-check the sequences of the CSP repeat regions provided in Supplementary Table 1. Especially the sequence for *P. vivax* CSP seems incorrect. According to me, this repeat region should be Arginine-rich, yet no Arg can be found in the provided sequence.

Comment 3: The introduction mentions the induction of a spiral form structure in CSP as a result of antibody binding. I propose to rephrase this sentence as there is no definitive proof for such a statement. According to me, it remains unclear whether the antibodies bind an existing spiral conformation or induce (and stabilize) it upon binding.

COMMENTS RELATED TO RESULTS

Comment 4: Page 4, small typo in the sentence "Phenotyping of CSP repeat mutants was always performed in parallel with a control:" The single colon should be a full stop.

Comment 5: Figure 1. Here, I have three comments/questions.

First, I would invite the authors to check if the positions of the molecular weight markers in Figure 1B are correct. The reason being that in our hands, (recombinant) *P. berghei* CSP always migrates with an apparent molecular mass higher than 50 kDa. Furthermore, in Suppl. Figure 4, the band migrates above 50 kDa in the left panel, and below 50 kDa in the lower panel, which seems inconsistent. In addition, the gels always show a double-band pattern for CSP, which is not discussed. May I ask the authors to elaborate?

Second, Figure 1D could benefit from an indication of the delineation of the salivary glands. While for experts in the field this may seem obvious, this is less so for non-experts.

Third, the legend mentions that antisera specific for the disordered region in the CSP C-terminus are employed. I propose to indicate/mention which region is specifically targeted.

Comment 6: Page 6 and Figure 3. Upon inspection of Figure 3A, it indeed becomes clear that the data points for Δ Rep2 and Scr are lower compared to controls as mentioned by the authors. However, upon comparison of Δ Rep2 with Scr it is obvious that the values for Δ Rep2 are much lower compared to those for Scr, yet this is not discussed by the authors. Do the authors have an explanation for this phenotype? Including this into the manuscript would be an added value. The legend of Figure 3 also contains a small typo: "sporozoite" should not be in bold.

Comment 7: Video S2. I invite the authors to double-check this file as this was the only video I could not play. While this could be related to settings on my computer, it may be worthwhile checking if the file is not corrupted in some way.

Comment 8: Figure 4. Here I propose to strengthen the link between the text and this figure even more by providing a definition for "meandering". While "circular", "patch" and "waving" are clearly defined and discussed in the text, "meandering" is not.

Comment 9: Page 8, small typo in the sentence "To better understand the motility defects in Δ Rep2 and Scr sporozoites, we analyzed these instantaneous speeds plots...". "Speeds plots" should be "speed plots".

Comment 10: Figure 7. The figure legend contains typos in the section describing panel (B). Here, the representative images are shown in the "upper panel" (and not "in (A)" as the text states now), and the quantitative analysis is shown in the "lower panel" (and not "in (B)" as the text states now).

Comment 11: I propose to dig deeper into the biophysical analysis of the single-molecular force spectroscopy data. I think it should be possible to extract values for the spring constants / elastic moduli, which should be reported (biophysics is a quantitative discipline after all). Furthermore, these could be compared to values of similar parameters reported for peptides corresponding to the *P. vivax* CSP repeat region (Kucharska et al., PMID 35023832). If such a comparison is meaningful, it could be of interest to include this in the manuscript as it may hint that the biophysical properties of the tandem repeat regions are conserved across CSPs from different Plasmodium species (even if the composition of the tandem repeats are different).

COMMENTS RELATED TO DISCUSSION

Comment 12: Page 15, first paragraph. I don't fully agree with this interpretation. While Δ Rep2 is indeed the most truncated mutant, the linker could still be long enough to enable the N-terminus to mask the C-terminus. The deleted part may also offer the linker certain conformational properties required for N to mask C. Hence, I do not agree that this can, considering the presented data, be interpreted as a pure "length requirement". Instead, I propose to interpret this as a "length and/or conformational requirement" in both the results and discussion section.

Referee #2:

This research Aims to understand the importance of the repeat regions of *P. berghei* CSP for sporozoite formation, function and activity. To this end, the three repeat regions of CSP were deleted and in addition, a further mutation was created that had scrambled repeats.

Results show that deletion of the extensive repeat 2, ultimately, has a profound effect on sporozoite motility. Similar phenotypes were observed for the scrambled mutant. However, deletion of the shorter repeat regions, termed one and three in this research, did not show an obvious phenotype.

The methodology used to dissect the mutant lines is exquisite and enabled great insight into the function of the central repeat 2 region of *P. berghei* CSP.

However, over all, there is a lack of nuance in the explanation of the data and the Figures are especially hard to follow and most need to be relabelled for ease of understanding. In addition, claims made in the text, are often not born out by the results shown in the Figures and vice versa. A clear example of this is Figure 4, where reference in panel B is made to a 'meandering' from of movement which is not shown in panel A and not mentioned at all in the text.

Overall, the lack of accuracy made the paper a challenge to read despite the fact that some really fascinating insights have been made by the study and I am wholly enthusiastic about the data.

I have made DETAILED edits and suggestions to the PDF of the submitted manuscript which I strongly encourage the authors to take into account should they choose to create a revised submission.

Referee #3:

Balaban et al describe an interesting and important study on the major malaria vaccine candidate CSP. They have generated mutants of the repeat region of CSP to elucidate its function, showing that the second repeat is important for sporozoite motility for salivary gland entry and liver stage infection, both central steps in transmission. The authors have assessed motility defects using a variety of elegant and controlled assays that show the repeat is important for parasite movement and these experiments are convincing. The study assesses peptides of CSP with respect to mechanical properties using a FRET-based assay, providing evidence for the repeat region having elastic properties suggesting it may be a spring. I find this evidence less compelling in the current version of the manuscript but declare that these experiments fall outside my direct area of expertise.

-Major defects in salivary gland numbers are reported. The authors indicate this is due to a salivary gland invasion issue but the evidence is not yet direct. Is it possible that motility in the midgut is an upstream issue? It remains possible that the defect originates in oocyst sporozoites being defective for motility. Can the authors measure with live imaging gliding before or soon after oocyst egress. Alternatively, the conclusion that the defect is explicitly at the salivary gland invasion step could be softened to account for additional possibilities, including the gliding defects shown in Fig 3.

-Page 4, 'side by side comparisons' for WT-GFP and RCon showed similar phenotypes. This is true, though some were reduced for the latter ie. Supp Fig. 2C, F. This could be indicated more clearly.

-Fig 1A. Adding R1, R2, R3 above each repeat would assist with clarity

-Fig 2A, Rep3 has three asterisks suggesting a statistical increase like Rep2, yet the text suggests this mutant has no C-terminal labelling.

-Fig 3. Were the Rep2 and Scr sporozoites used derived from the salivary glands or the hemolymph (eg. see blots in Fig 1B)? As this location affects sporozoite infectivity for the mammalian host, it would be helpful to make this clear in the text or figure legend. Also, combined results of EEF number and size in Fig 3B - it may be easier to interpret in 2 separate graphs. It took me some time to decode that the second group was referring to size only, not a repeat experiment.

-Fig 4B, please indicate units for the y-axis. Has 'meandering' been defined previously? If so please cite; if not, please include an example schematic and video. Does Rep3 engage in significantly 'less' patch gliding than control?

-Fig 5A, please supply appropriate parametric statistical analyses to demonstrate significant differences for Rep2 and Scr.

-Please indicate which 'Control' parasite line (WT-GFP or RCon) was used in each experiment.

-Fig 7B. The authors suggest that TRAP is still secreted to the surface of Rep2 and Scr sporozoites but not processed from the parasite membrane into trails (p10). What is the evidence for secretion to the surface? The microscopy images supplied in Fig. 7B do not convince me of that conclusion, and while not critical to this story, a quantitative analysis would be required to draw that conclusion. TRAP may remain on the surface as the small proportion of sporozoites that do move may not do so properly. I assume, though there is no reference to it in the text, that the rhomboid cleavage site in CSP is mapped, that it is near the C-terminus, and that it is still present in the mutants? It would be helpful to show the sporozoite in each image. For example, if the brightfield images or GFP are available or another marker (eg see Fig. 7C).

-The data in Fig. 8 are interesting indeed. While I am not a detailed expert in this type of analysis, it seems to me that a previously validated spring peptide control would be helpful to validate the assay. The overlapping stretch/relax curves for WT CSP peptide in Fig. 8A are convincing. While reading the Grashoff et al reference, which also used Cy3 and Cy5 fluorophores, the curves were most sensitive (exponential) between 1-6 pN force (their Fig. 2i, S4b) indicating these fluorophores are most sensitive within this range (probably judicious not to extrapolate too far beyond this?). The data in Fig 8 with CSP peptides does not appear to show this same sensitivity curve for the same fluorophores (and also presents data out to 25-30 pN, potentially above the most sensitive range reported previously). It may be challenging, but can the authors deduce why their assay with Cy3/5 does not show a similar sensitivity range? Does this affect the analysis or the interpretation? I suggest including a control

spring peptide(s) that can reassure the assay is working as expected in their hands, and a statistical analysis showing hysteresis of Scr peptide is due to curves truly not overlapping (eg, see noise in Fig 8C is also not clearly interpretable?), to bolster the important conclusion that CSP repeats indeed form a spring.

-Supp Fig. 1 Please indicate what NTC is

-Supp Fig. 4 Please indicate on the blot which band is the cleaved CSP species and which is uncleaved. I understand the cleaved species to be the very faint lower band in each lane. If so, this is a challenging result to conclusively interpret. Such intensity measurements of uncleaved and cleaved bands may not be within the linear range at the same time for quantification; different exposure times are likely needed to confirm this, however, shorter times may preclude detection of the fainter cleaved band. In the current blot, the upper band may have reached saturation (beyond the linear range) while the lower band is now just detectable (and within the linear range). Therefore this approach, elegant as it is, should be interpreted with caution. The amount of Rep2 is, in my mind, too close to call from one blot with one exposure time. Additional repeats may help to confirm that Rep2 is truly as abundant, or less, than control, but I acknowledge these 35-S-Met experiments are not trivial (I greatly appreciate them!). One can soundly conclude that processing was not inhibited. But how does this blot answer whether 'premature processing' occurred in Rep2, as stated in the text (p6)? Was a pulse-chase conducted?

We thank all of the reviewers for the helpful and insightful reviews. Below are our responses, in red, which in many cases led to changes in the paper that have improved the manuscript.

Referee #1:

The paper is well written and the results are presented in a logical fashion. The experiments seem technically well executed and the data appear to be carefully analyzed. The figures are clear and the provided movies are an absolute added value to the manuscript. The presented findings are novel and relevant. While most manuscripts in the literature have focused on specific CSP domains or regions, studies on the overall architecture of the full-length protein are scarce (despite decades of research). The paper provides valuable insights into the functional role of CSP's molecular organization under native conditions at the parasite surface. The choice for *P. berghei* CSP is logical as this represents a tractable model. The findings obtained for *P. berghei* CSP are of additional interest as some of these molecular principles could apply to the functional architectures of CSPs from human malaria parasites (e.g., *P. falciparum* and *P. vivax*). My suggestions to adapt the manuscript are related to changes in the text and to dig deeper into the existing data sets. This reviewer sees no need for additional wet lab experiments.

We thank the Reviewer for their helpful and insightful comments which we address below and in the text.

COMMENTS RELATED TO THE INTRODUCTION

Comment 1: The introduction is well written. However, it is this reviewer's opinion that it lacks information on the structural features of CSP domains that have been thoroughly characterized in the past. As this journal has a relatively broad readership (including scientists that are not active within the malaria/CSP field), I think this could be of interest to the reader and therefore propose to provide more information on this topic. For example, the CSP C-terminal and N-terminal domains have been thoroughly structurally characterized by Doud et al. (PMID 22547819) and Geens et al. (PMID 38059674), respectively.

The reviewer makes an excellent point and this information with the references above, have now been added to the Introduction in the first paragraph on page 3.

Comment 2: I kindly ask the authors to double-check the sequences of the CSP repeat regions provided in Supplementary Table 1. Especially the sequence for *P. vivax* CSP seems incorrect. According to me, this repeat region should be Arginine-rich, yet no Arg can be found in the provided sequence.

Thank-you for pointing this out. Actually many of the sequences were inaccurate because they were taken from an old paper (Kemp D et al., *Annu Rev Microbiol* 41, 1987). We have now obtained all sequences from the most recent version of our genome database (plasmodb.org) and revised the Supplementary Table accordingly.

Comment 3: The introduction mentions the induction of a spiral form structure in CSP as a result of antibody binding. I propose to rephrase this sentence as there is no definitive proof for such a statement. According to me, it remains unclear whether the antibodies bind an existing spiral conformation or induce (and stabilize) it upon binding.

While our interpretation of the two papers by Oyen et al., (PMID: 29138320; PMID: 29138320) was that the Fab portion of the antibody binds to itself creating the spiral, we are not experts in this area and defer to the reviewer. We have rephrased the sentence as follows:

Additionally, studies with recombinant CSP bound to repeat-specific antibodies showed that the repeats can take on a spiral structure, though this may be induced or stabilized by interaction with antibody. Page 4 first paragraph

COMMENTS RELATED TO RESULTS

Comment 4: Page 4, small typo in the sentence "Phenotyping of CSP repeat mutants was always performed in parallel with a control:" The single colon should be a full stop.

corrected

Comment 5: Figure 1. Here, I have three comments/questions.

First, I would invite the authors to check if the positions of the molecular weight markers in Figure 1B are correct. The reason being that in our hands, (recombinant) P. berghei CSP always migrates with an apparent molecular mass higher than 50 kDa. Furthermore, in Suppl. Figure 4, the band migrates above 50 kDa in the left panel, and below 50 kDa in the lower panel, which seems inconsistent. In addition, the gels always show a double-band pattern for CSP, which is not discussed. May I ask the authors to elaborate?

Thank-you for raising this point. In Figure 1B, the MW markers are correct. Though somewhat faint, the full-length CSP band is visible above the darker cleaved band and migrates with an apparent molecular mass over 50 kDa. We have found that the relative intensity in western blots of the full-length CSP and the cleaved form varies according to dissection time, with longer dissection times leading to more cleaved form than full-length. This was the case here because we dissected 3 different parasite lines for each western. We have now included in the text an explanation of the double band pattern, i.e. cleaved and uncleaved CSP which has long been observed:

"By western blot CSP is generally observed as two bands, a full-length form and a cleaved form (Coppi et al., 2005; Yoshida et al., 1981; Cochrane et al., 1982)."

For the metabolic labeling experiments, the lack of MW markers on the second gel was an oversight and we have now added this to the second gel in Supplemental Figure 4A. Initially only full-length is labeled but as the labeling goes beyond ~45 minutes, we begin to see the cleaved form. We have also labeled both forms of CSP in each gel and have added this information to the legend of Supp Fig 4:

"Metabolic label is initially incorporated into full-length CSP (top band) and over time a small amount of cleaved CSP can be observed (faint bottom band)."

Second, Figure 1D could benefit from an indication of the delineation of the salivary glands. While for experts in the field this may seem obvious, this is less so for non-experts.

Good point and now done.

Third, the legend mentions that antisera specific for the disordered region in the CSP C-terminus are employed. I propose to indicate/mention which region is specifically targeted.

This information was included in the Methods section under "Antibodies" and now also included in the figure legend.

Comment 6: Page 6 and Figure 3. Upon inspection of Figure 3A, it indeed becomes clear that the data

points for Δ Rep2 and Scr are lower compared to controls as mentioned by the authors. However, upon comparison of Δ Rep2 with Scr it is obvious that the values for Δ Rep2 are much lower compared to those for Scr, yet this is not discussed by the authors. Do the authors have an explanation for this phenotype? Including this into the manuscript would be an added value.

The legend of Figure 3 also contains a small typo: "sporozoite" should not be in bold.

Typo corrected. We think that the stronger DeltaRep2 phenotype is due to the fact that it has two issues, first the repeat's function of a linker that leads to masking of the TSR is not occurring properly such that salivary gland entry is impaired and second, the function of the repeats as a surface on which adhesion sites can form is also impaired due to the severe truncation. We have now added a paragraph to the Discussion to explain this and agree that this is an important addition to the analysis of our results.

Comment 7: Video S2. I invite the authors to double-check this file as this was the only video I could not play. While this could be related to settings on my computer, it may be worthwhile checking if the file is not corrupted in some way.

Thank-you for pointing this out. Video S2 is now an MP4 and should be able to be opened on all computers.

Comment 8: Figure 4. Here I propose to strengthen the link between the text and this figure even more by providing a definition for "meandering". While "circular", "patch" and "waving" are clearly defined and discussed in the text, "meandering" is not.

This was an oversight which we have now addressed. We have now added this to the text (page 8, 1st paragraph), with a new reference as well as adding an explanatory sentence to the figure legend and in the Results section. See modified text below:

We categorized sporozoite motility according to previously observed motility phenotypes: continuous circular gliding, patch gliding, attached waving, or meandering (Vanderberg et al., 1974, Hegge et al., 2010; PMID: 3054075). Examples of circular gliding, patch gliding, and attached waving are shown in Figure 4A and in Videos S1-3. Circular gliding, in which sporozoites move in continuous circles, is the predominant form of motility observed in wild-type parasites and is an indicator of fully mature and infectious sporozoites (Video S1). Meandering is a form of circular gliding in which the sporozoites move in longer, more open arcs rather than tight circles (Stewart et al., 1988).

Comment 9: Page 8, small typo in the sentence "To better understand the motility defects in Δ Rep2 and Scr sporozoites, we analyzed these instantaneous speeds plots...". "Speeds plots" should be "speed plots".

corrected

Comment 10: Figure 7. The figure legend contains typos in the section describing panel (B). Here, the representative images are shown in the "upper panel" (and not "in (A)" as the text states now), and the quantitative analysis is shown in the "lower panel" (and not "in (B)" as the text states now).

We have now added additional panel labels so this should be corrected.

Comment 11: I propose to dig deeper into the biophysical analysis of the single-molecular force spectroscopy data. I think it should be possible to extract values for the spring constants / elastic moduli,

which should be reported (biophysics is a quantitative discipline after all). Furthermore, these could be compared to values of similar parameters reported for peptides corresponding to the *P. vivax* CSP repeat region (Kucharska et al., PMID 35023832). If such a comparison is meaningful, it could be of interest to include this in the manuscript as it may hint that the biophysical properties of the tandem repeat regions are conserved across CSPs from different *Plasmodium* species (even if the composition of the tandem repeats are different).

We now report the compliance values and peptide-length normalized values in the revised manuscript (see Results, page 13, 1st paragraph). In addition, we calculated elastic modulus from our own data and compared it to the published value from *P. vivax* CSP repeat region, and discuss a possible explanation for the discrepancy (see Results, page 13, 2nd paragraph).

COMMENTS RELATED TO DISCUSSION

Comment 12: Page 15, first paragraph. I don't fully agree with this interpretation. While $\Delta\text{Rep}2$ is indeed the most truncated mutant, the linker could still be long enough to enable the N-terminus to mask the C-terminus. The deleted part may also offer the linker certain conformational properties required for N to mask C. Hence, I do not agree that this can, considering the presented data, be interpreted as a pure "length requirement". Instead, I propose to interpret this as a "length and/or conformational requirement" in both the results and discussion section.

While we stated that it's a length requirement b'c of the C-terminal exposure, conformation is also a possibility and that has been added, see modified text below:

Since the TSR domain is normally masked in salivary gland sporozoites (Coppi et al., 2011, Hopp et al., 2015), our data suggests that there is a length and/or conformational requirement for masking of the TSR. (Page 17, 1st paragraph)

Referee #2:

The methodology used to dissect the mutant lines is exquisite and enabled great insight into the function of the central repeat 2 region of *P. berghei* CSP.

However, over all, there is a lack of nuance in the explanation of the data and the Figures are especially hard to follow and most need to be relabelled for ease of understanding. In addition, claims made in the text, are often not born out by the results shown in the Figures and vice versa. A clear example of this is Figure 4, where reference in panel B is made to a 'meandering' from of movement which is not shown in panel A and not mentioned at all in the text.

Overall, the lack of accuracy made the paper a challenge to read despite the fact that some really fascinating insights have been made by the study and I am wholly enthusiastic about the data.

I have made DETAILED edits and suggestions to the PDF of the submitted manuscript which I strongly encourage the authors to take into account should they choose to create a revised submission.

We thank Reviewer 2 for constructive comments. We have addressed the concern mentioned above and the comments highlighted in the pdf as outlined below:

Meandering in Figure 4 – This was an oversight that we have now addressed in the Results section and in the figure legend, with a new reference added (for more detail see response to the same issue in the response to Reviewer 1).

Title: While we make it very clear in the paper that our studies are on *Plasmodium berghei*, given the highly conserved features of the repeats among all *Plasmodium* species, we believe that *P. berghei* is a model for studying CSP that has proven over and over again to be faithful to the human malaria parasites. Indeed, all of the proof of concept work on the CSP repeats as a vaccine candidate came from the rodent model and served as the basis for what are now the only two licensed malaria vaccines (RTS,S and R21). Thus, we prefer to keep the title as is because it opens up the relevance of this work to CSP repeats from other species. Indeed this similar to what has been done in previous publications in EMBO journals (PMID: 31368598 and PMID: 33666362, PMID:11927544).

Abstract: How can a spring be both stiff and elastic? Elasticity, which is defined in the Results section (pg. 11), refers to the ability of a protein/peptide to unfold and fold without energy loss or protein rupture whereas the stiffness refers to the force needed to stretch or deform a protein/peptide. So while it can take more force to stretch the CSP repeats compared to spider silk, they are still elastic in that they can unfold and fold without hysteresis. We have added this in the Results section (page 13, 1st paragraph) as space does not allow us to add to the Abstract:

“Thus while like the spider silk protein, the CSP repeat peptides are elastic in that they can unfold and fold without hysteresis, higher force is required to stretch the CSP repeat peptides compared to the spider silk peptide.”

Significance Statement: sentence changed to be less awkward.

You refer to Figure 1B after Figures 1C and 1D - please correct the order – We struggled to change the order of the figures as we agree that ideally 1B should be discussed before 1C&D. The issue is that 1D does not fit into a slot next to 1C and there is not sufficient space to have a figure below the current 1D. Though we could shift 1B and 1C, that would not solve the problem and might make it more confusing since 1C and 1D are discussed together.

please change Cdis to CSP-Cdis for ease of interpretation – This has now been changed in the Figure, and more detail has been added to the Figure legend and Results section.

Statistics in 2A – We have now added a sentence that includes the statistically significant difference between control and Rep3. The increased C-terminal labeling of Rep3 was not as high as we observed for Rep2 and we have also added statistics showing the significant difference between these two mutants to the figure and discuss this in the Results, (page 6, 2nd paragraph).

Figure 5 panels – We have now inserted more precise labeling of the panels in Figure 5, with concordant changes to the text and legend.

Figure 6 panels – Agree that it makes more sense to start with circular gliding so we’ve changed the figure as suggested and also added an additional panel label, changed from “circle” to “circular” and made the necessary adjustments to the text and figure legend.

“rear” – changed to “posterior”

Figure 6 - statistics - no **** above delta Rep2 for turnover – Agree and we have now added some explanation to the sentence as follows: *"When we quantified these events, we found fewer adhesion site turnover events per unit time in Δ Rep2, and Scr mutants though we had fewer datapoints for Δ Rep2 and it did not reach statistical significance"* (page 10, 1st full paragraph).

Figure 7 - The images are not specifically referred to – now done and additional panel designations added.

The Scr image appears to be cut off on the bottom right – now corrected.

Also, there are no scale bars in the images – now corrected.

Since the actively moving sporozoite is so motile, its hard to see the lack of TRAP on its surface but I assume the sporozoite is top right – the sporozoite was actually not visible since very little TRAP is on the surface compared to the trail. We've now made a new figure panel (Fig 7B) to include both fluorescent and phase images so that one can see the location of the sporozoite.

Since the Scr and DeltaRep2 sporozoites don't move, TRAP can't be secreted, right – Actually it can be and is secreted but its not shed. We agree that this needed some clarification and this is now explicitly stated in this paragraph as follows: *"In contrast, we found that Δ Rep2 and Scr sporozoites had a marked increase in detectable TRAP on their surface, indicating that it is secreted onto the surface, but TRAP was not detected in trails, suggesting it is not able to be shed"* (end of page 10 and beginning of page 11).

Change the work 'patch' - Though we understand that both uses of the word "patch" might be confusing, this is supported by the literature (Swearingen et al., 2016; Munter et al., 2009), and to maintain consistency with the literature, we thought it best to stick with the previously codified terms.

is a dot a small patch - how did you delineate a dot from a patch? – We have now added some detail to the legend of Figure 7F to better explain this.

Is MTIP area per sporozoite – yes and we have now clarified in the legend

In this section, you only look at the NANDPAPP repeat. you do not study the NPNDPPPP or the PQPQPRPO repeat. I suggest you do this too and if you are not able to do this, you need to talk about why – We completely agree that this would be interesting, as would studying the repeat peptides of other species. However, after the generation and phenotyping of the mutants it became clear that it was the repetitive structure and length of the repeats that were important and so that is where the single-force microscopy experiments were focused. We now state the reason why we focused on the NANDPAPP repeat in the 2nd paragraph of this section:

"Because phenotyping of the CSP repeat mutants suggested the repetitive structure and the length of the repeats were critical to their function, we tested different length repeat peptides and a scrambled repeat peptide" (page 12, 1st full paragraph).

Also, you should also discuss in this section, based on your results and previous results, why you think the delta Rep2 parasite is so unfit - i assume because it lacks the specific NANDPAPP???? – Thank-you for pointing this out and we have now added a paragraph to the Discussion (page 14, 2nd paragraph) to explain this as we agree it is an important point. We think that the DeltaRep2 phenotype is due to the fact that it has two issues, first the repeat's function as a linker is not occurring properly so there is premature exposure of the TSR leading to sporozoites that do not efficiently enter salivary glands because they are adhering to other mosquito organs (see Coppi et al., 2011 for this phenotype in the DeltaN CSP sporozoites). In addition, as our gliding motility and RICM adhesion site experiments indicate, the length of the repeats is important for proper adhesion site formation and turnover on the sporozoite surface. Thus, while the Scr sporozoites only have a defect in motility, i.e. the CSP no longer serves as a proper milieu for TRAP-containing

adhesion sites to form, the DeltaRep2 has two additive phenotypes, both the above and its function as a linker.

I don't think that Figure 8A shows what you suggest in this sentence - please change the Figure panel or provide a better explanation.

We updated the figure and figure legend to avoid confusion and now show how increasing the separation between Cy3 and Cy5 with the application of force stretches the peptide and thus, leads to a change in the FRET signal.

why would FRET efficiency be lower due to increased peptide length - Different lengths of CSP repeats (CSP₂₄ and CSP₄₀) were sandwiched between identical dsDNA handles harboring the donor and acceptor dyes. As FRET efficiency is a proxy for end-to-end distance of the peptide, and scales inversely with the separation between the dyes, longer the peptide (CSP₄₀), lower the FRET efficiency. We hope that the revised figure is helpful in this regard and have also revised the sentence to avoid confusion:

"A similar elastic behavior without hysteresis was also observed for a longer 40 amino acid CSP repeat peptide (NANDPAPP)₅, referred to as CSP₄₀ (Figure 8C) but the FRET efficiencies are lower across all force values compared to CSP₂₄ (Figure 8B) due to the increased peptide length" (page 12, last paragraph).

Also, as before, have B, C D and E for the four panels and label appropriately. B = CSP₂₄, C = scrambled, D = CSP₄₀ and E = as labeled – Done

i strongly encourage you to clearly delineate the repeat mutant phenotypes at the beginning of your discussion. deletion of repeat blocks 1 and 3 have apparently no effect on sporozoite activity. and yet, you claim that the repeats are essential to sporozoite function. you need to be VERY specific in your wording. the attention to detail and nuance in your discussion is disappointing

We agree and now address this in the new 2nd paragraph of the Discussion.

Regarding the following statement: *"The repeat mutants generated in the current study develop normally in the oocyst, indicating that the mutations we introduced do not lead to gross misfolding of CSP and allowing for downstream functional analysis in salivary gland sporozoites"* - this is not a true statement. deletion of repeat 2 has a profound effect on sporozoite residency in the salivary gland. indeed, you claim that certain analyses were not possible as you were unable to isolate sufficient numbers of sporozoites for testing.

We respectfully disagree with this assessment. The key phenotype that we avoided with our mutants was a developmental one in the oocysts. Since CSP is essential for sporozoite formation (PMID: 9002517), many CSP mutations lead to a complete or partial defect in sporozoite formation in oocysts, completely precluding any downstream analysis. Investigators have tried extra copies under different promoters, conditional systems, etc to see downstream functions of CSP, many of which are unpublished because sporozoite development in oocysts is exquisitely sensitive to the timing, conformation and amounts of CSP (PMID: 25438048; PMID: 11927543; PMID: 33600048; PMID: 38396332). Our data demonstrate normal sporozoite development in oocysts and normal exit from oocysts, thus avoiding the dreaded oocyst phenotype. While its true that DeltaRep2 had a profound effect on sporozoite entry into salivary glands, from our vantage point, this is a downstream phenotype and we were able to obtain sufficient numbers of salivary gland sporozoites to perform the vast majority of the assays in the paper (Figs 1, 2, 3A&D, 4, 5, 6 & 7).

please refine Figure 9 to clearly show a difference between wildtype CSP and mutant CSP. arrows indicating movement (or lack thereof) of the adhesion sites would be most beneficial. The two cartoons are now labeled. We can be fairly certain of the difference in the size of the adhesion site from our data but we are still not able to live image CSP or TRAP on the sporozoite surface during gliding so we thought rather than arrows showing

movement of the sporozoite surface, we would indicate that the difference is in adhesion site size and thus, the size of the attached actin filament which would be expected to lead to motility defects. This is now added to the legend.

the rep 1 and 3 are also different to rep 2 as well as being N and C terminus to rep 2 and their deletion appears inconsequential. is it simply a numbers game?? if one were to replace rep 2 with rep 1, would the sporozoite be functional??? if one deleted half of rep 2, would the sporozoite be functional - there is ample breadth for discussion here and you could provide the reader with hypotheses for further testing.

We agree that this should have been discussed and have now added this to the new paragraph 2 in the Discussion. Indeed our interpretation is that it is a “numbers game” and we completely agree that more mutants, including the ones you mention above, would be informative. Similar to performing additional single molecule studies with other peptides, other mutants are an area of future investigation and this is included in the aforementioned paragraph.

this is 9A and you refer to it AFTER 9B. In addition, were you able to take images of this type of motion for delta rep 1 and delta rep 3 and were they the same??? i realize that delta rep 2 would obviously not show the same pattern. A needs a scale bar and a methodological explanation.

We have now changed the order of the Figure 9 panels and a scale bar has been added. Unfortunately we could not perform the in vitro 3-D assay with our mutants because the GFP they express is under a weak promoter (so as not to interfere with protein expression levels) and they could not be tracked in 3D as the matrix dims the signal significantly. A scale bar has been added to figure and methods added to the Methods section under Live Gliding Assays.

Referee #3:

Balaban et al describe an interesting and important study on the major malaria vaccine candidate CSP. They have generated mutants of the repeat region of CSP to elucidate its function, showing that the second repeat is important for sporozoite motility for salivary gland entry and liver stage infection, both central steps in transmission. The authors have assessed motility defects using a variety of elegant and controlled assays that show the repeat is important for parasite movement and these experiments are convincing. The study assesses peptides of CSP with respect to mechanical properties using a FRET-based assay, providing evidence for the repeat region having elastic properties suggesting it may be a spring. I find this evidence less compelling in the current version of the manuscript but declare that these experiments fall outside my direct area of expertise.

We thank the Reviewer for their insightful comments which have improved the manuscript.

-Major defects in salivary gland numbers are reported. The authors indicate this is due to a salivary gland invasion issue but the evidence is not yet direct. Is it possible that motility in the midgut is an upstream issue? It remains possible that the defect originates in oocyst sporozoites being defective for motility. Can the authors measure with live imaging gliding before or soon after oocyst egress. Alternatively, the conclusion that the defect is explicitly at the salivary gland invasion step could be softened to account for additional possibilities, including the gliding defects shown in Fig 3.

The reviewer raises an important point that we felt we had addressed with hemolymph sporozoite counts shown in Supplemental Figure 3 where we counted oocyst and hemolymph sporozoites in 4 to 5 independent mosquito cycles of control and mutant lines and found no differences in hemolymph

sporozoite counts between these lines. These data indicate that sporozoites exit oocysts and enter the open circulatory system of the mosquito normally. Data from collaborator Dr. Frischknecht has shown that motility is required for sporozoite egress from oocysts and that this is a different type of motility than the gliding motility necessary for salivary gland invasion (PMID: 28115054). The latter point has also been demonstrated by TRAP KO parasites which exit oocysts normally but do not enter salivary glands due to their gliding motility defect. To make sure this point is clear we have added some text to the following sentence:

“We found that these mutants had similar numbers of hemolymph sporozoites as controls indicating that they exit oocysts normally (Supplementary Figure 3)” (page 5, last paragraph).

-Page 4, 'side by side comparisons' for WT-GFP and RCon showed similar phenotypes. This is true, though some were reduced for the latter ie. Supp Fig. 2C, F. This could be indicated more clearly.

We have edited the sentence to reflect this: *“Side by side comparisons of WT-GFP and RCon parasites showed that they had similar phenotypes in the assays used in our study though RCon sporozoites had slightly reduced maximum speed and time with multiple adhesion sites”* (page 5, 1st paragraph).

-Fig 1A. Adding R1, R2, R3 above each repeat would assist with clarity

While we agree that if the schematic was a stand-alone figure this would be an excellent idea, the current Figure 1 is packed and when we added the Rep 1 Rep 2 Rep 3 labels it was very crowded and not really easier to interpret. Note that we cannot do R1, R2, R3 because of Region I which is RI. Since we are already at the Figure maximum, we could put Fig 1A into the Supplemental. We discussed this and felt that it was important to have in the main body but if you think we should move it, we can. Currently the color coded repeat sequence matches Rep 1, Rep 2, and Rep 3 and this is explained in the legend. We have edited the sentence in the legend to make this more clear: *“The three repeat sequences are color coded in both the sequence and CSP schematic: orange = first repeat, blue = second repeat, and purple = third repeat”*

-Fig 2A, Rep3 has three asterisks suggesting a statistical increase like Rep2, yet the text suggests this mutant has no C-terminal labelling.

We have now added a sentence that includes the statistically significant difference between control and Rep3. The increased C-terminal labeling of Rep3 was not as high as we observed for Rep2 and we have also added statistics showing the significant difference between these two mutants to the figure and discuss this in the Results (page 6, 2nd paragraph).

-Fig 3. Were the Rep2 and Scr sporozoites used derived from the salivary glands or the hemolymph (eg. see blots in Fig 1B)? As this location affects sporozoite infectivity for the mammalian host, it would be helpful to make this clear in the text or figure legend.

The only experiment in which we used hemolymph sporozoites was the western blot shown in Figure 1. This was because we could not get clean westerns with salivary gland sporozoites of DeltaRep2 and Scr mutants due to the higher number of salivary glands we had to dissect to obtain the sporozoites, and thus the significantly increased amount of mosquito material. We've added the following sentence at the end of this paragraph: *“Of note, these were the only experiments in this study for which we used hemolymph sporozoites”* (page 6, 1st paragraph) and added the “salivary gland” modifier in several places to make this clear.

Also, combined results of EEF number and size in Fig 3B - it may be easier to interpret in 2 separate graphs. It took me some time to decode that the second group was referring to size only, not a repeat experiment.

Now split into two graphs

-Fig 4B, please indicate units for the y-axis. Has 'meandering' been defined previously? If so please cite; if not, please include an example schematic and video. Does Rep3 engage in significantly 'less' patch gliding than control?

Regarding the Y-axis, the graphs in 4B were made in Prism as “parts of a whole” and don't have a Y-axis. Thus, we feel uncomfortable fashioning our own y-axis. Importantly, the message of the graphs is pretty clear and this is confirmed by the statistics we performed, which are described in the legend.

We have now included an explanation of “meandering” in the text (with a citation) and in the figure legend.

The only statistically significant differences were those mentioned in the figure legend – no significant difference in patch gliding between DeltaRep3 and Control.

-Fig 5A, please supply appropriate parametric statistical analyses to demonstrate significant differences for Rep2 and Scr.

Panels A-C are for illustrative purposes, summarizing the frequency distribution of speeds within and across groups. Thus, they give the reader an idea of the speed distributions for each parasite line. However, since speed at sequential timepoints are not independent of one another, i.e. sporozoites move by an adhesion based motility so that they go fast (when adhesion sites are released) and then slower (when new adhesion sites are formed) the sequential speeds are correlated with one another making statistical comparisons of the frequency plots difficult. In consultation with a biostatistician at Hopkins, we determined that the best way to perform statistics on these data was to compare median and maximum speeds of individual sporozoites from each line. This allows for comparison across the different lines, and is a way of demonstrating the statistically-significant differences among the different parasite lines that are suggested by the data in panels A-C. These data are shown in panels D and E. Clearly there are additional ways to compare them but we felt that median and maximum speeds were the most relevant.

-Please indicate which 'Control' parasite line (WT-GFP or RCon) was used in each experiment.

The data shown for each experiment is pooled data from 3 to 8 independent mosquito cycles. For each mosquito cycle we usually had 3 cages of mosquitoes, 2 mutants and 1 control, with the control being either WT-GFP or RCon because RCon was not successfully generated until about mid-way through the project. Thus for many of the experiments we used both WT-GFP and RCon controls and those data were pooled as were the mutant data from the different mosquito cycles. We have revised the sentence to be more clear about this:

“Thus, throughout the manuscript, “Controls” refer to either WT-GFP, RCon or a combination of both lines” (1st paragraph page 5).

-Fig 7B. The authors suggest that TRAP is still secreted to the surface of Rep2 and Scr sporozoites but not processed from the parasite membrane into trails (p10). What is the evidence for secretion to the surface? The microscopy images supplied in Fig. 7B do not convince me of that conclusion, and while not critical to this story, a quantitative analysis would be required to draw that conclusion.

The evidence for surface secretion are the IFAs of gliding sporozoites in Fig 7B-D. These are paraformaldehyde-fixed but not permeabilized sporozoites. When sporozoites are permeabilized a very

different staining pattern is obtained, namely a large part of the cytoplasm is stained because of the large number of TRAP-containing micronemes and the nucleus is spared (PMID: **10508153** PMID: **10816526**). We only see surface staining when sporozoites start to move or in the case of these and other mutants, try to move (see Fig 2D&E and Fig 5C&D of PMID: 22911675 where we create mutants that can secrete but not shed TRAP). Though this is discussed in the Discussion, we have tried to make this more clear in the Results section, with the following addition to the 2nd full paragraph on page 10: *“Previous immunofluorescence assays of permeabilized sporozoites have shown that TRAP is stored in specialized secretory vesicles called micronemes in a nuclear-sparing pattern (PMID: **10508153** PMID: **10816526**) and that when stimulated to start moving, it is secreted onto the sporozoite surface and shed by the action of a parasite rhomboid protease into the trails (PMID: **10816526**; Ejigiri et al., 2012). In mutants that are unable to move, secretion still occurs but shedding into trails does not such that significantly more TRAP is observed on the sporozoite surface (Ejigiri et al., 2012).”*

We quantify surface TRAP in Figure 7C, with methodology that does not permeabilize the sporozoite (described in the legend and the Methods section).

TRAP may remain on the surface as the small proportion of sporozoites that do move may not do so properly. I assume, though there is no reference to it in the text, that the rhomboid cleavage site in CSP is mapped, that it is near the C-terminus, and that it is still present in the mutants? It would be helpful to show the sporozoite in each image. For example, if the brightfield images or GFP are available or another marker (eg see Fig. 7C).

We agree and apologize for the oversight of not including sporozoite images next to the fluorescence images. This is now corrected. There is no reference to the precise mapping of the rhomboid cleavage site because it has not been done, however, we do reference and state in the paper that the rhomboid cleavage site is localized to the transmembrane domain of TRAP (Ejigiri et al., 2012).

-The data in Fig. 8 are interesting indeed. While I am not a detailed expert in this type of analysis, it seems to me that a previously validated spring peptide control would be helpful to validate the assay. The overlapping stretch/relax curves for WT CSP peptide in Fig. 8A are convincing. While reading the Grashoff et al reference, which also used Cy3 and Cy5 fluorophores, the curves were most sensitive (exponential) between 1-6 pN force (their Fig. 2i, S4b) indicating these fluorophores are most sensitive within this range (probably judicious not to extrapolate too far beyond this?). The data in Fig 8 with CSP peptides does not appear to show this same sensitivity curve for the same fluorophores (and also presents data out to 25-30 pN, potentially above the most sensitive range reported previously). It may be challenging, but can the authors deduce why their assay with Cy3/5 does not show a similar sensitivity range? Does this affect the analysis or the interpretation? I suggest including a control spring peptide(s) that can reassure the assay is working as expected in their hands, and a statistical analysis showing hysteresis of Scr peptide is due to curves truly not overlapping (eg, see noise in Fig 8C is also not clearly interpretable?), to bolster the important conclusion that CSP repeats indeed form a spring.

We performed the same experiment and analysis that we previously reported for the spider silk peptide (Grashoff et al, PMID:20613844; Brenner et al, PMID:26821490), and found that, as reported in the revised text, the CSP peptide repeat is much stiffer (peptide-length normalized compliance is 0.002 nm/pN/a.a. for CSP₂₄ and CSP₄₀ compared to 0.012 nm/pN/a.a. for the spider silk peptide. As a result, it takes much higher forces to stretch the CSP peptides, explaining the observation noted by the reviewer. In Grashoff et al, the peptide was stretched beyond FRET-measurable distances because of the higher compliance. In contrast, the CPS peptides do not stretch beyond FRET range even at the highest forces

available in our instrument (between 25 and 30 pN). A control for noise amplitude is our data for the wild type CSP peptides, shown in Figure 8B and 8C where stretching and relaxation curves fully overlap within noise. In response to this comment, we have revised the text to present compliance values for both CSP₂₄ and CSP₄₀, their peptide-length normalized values, and compared them to those of spider silk peptide. We have modified the text in the Results section and added a paragraph to the Results to address these points (2nd paragraph page 13).

-Supp Fig. 1 Please indicate what NTC is
Now added to legend

-Supp Fig. 4 Please indicate on the blot which band is the cleaved CSP species and which is uncleaved. I understand the cleaved species to be the very faint lower band in each lane. If so, this is a challenging result to conclusively interpret. Such intensity measurements of uncleaved and cleaved bands may not be within the linear range at the same time for quantification; different exposure times are likely needed to confirm this, however, shorter times may preclude detection of the fainter cleaved band. In the current blot, the upper band may have reached saturation (beyond the linear range) while the lower band is now just detectable (and within the linear range). Therefore this approach, elegant as it is, should be interpreted with caution.

We now indicate full-length and cleaved forms of CSP in the gel photos and you are correct in that the cleaved species is the very faint lower band in each lane. While we think the identification of full-length and cleaved forms is pretty clear, particularly given all of our previous work in this space, we do agree that any quantification of cleavage efficiency in these experiments is challenging, particularly because we did not obtain sufficient numbers of DeltaRep2 sporozoites for a pulse-chase. This experiment was performed to determine whether the significantly increased exposure of the TSR domain in DeltaRep2 sporozoites that we observed in Figure 2 was because CSP was prematurely cleaved or whether the significant truncation of the repeats in DeltaRep2 led to less masking of the C-terminus. Since initial incorporation of the label, which would be in full-length CSP as it is being translated, is similar in DeltaRep2, it suggests (but does not prove) that CSP synthesis of full-length CSP in the DeltaRep2 is normal and thus premature cleavage is not a major issue with this mutant. Nonetheless, as you point out, the methodology is not sufficiently sensitive to say anything about the rate of cleavage, an opinion with which we completely agree. Though we only conclude that there is no difference in full-length DeltaRep2 CSP versus wildtype CSP, we have modified the text to emphasize that we do not provide definitive proof of this (see response below).

The amount of Rep2 is, in my mind, too close to call from one blot with one exposure time. Additional repeats may help to confirm that Rep2 is truly as abundant, or less, than control, but I acknowledge these 35-S-Met experiments are not trivial (I greatly appreciate them!). One can soundly conclude that processing was not inhibited. But how does this blot answer whether 'premature processing' occurred in Rep2, as stated in the text (p6)? Was a pulse-chase conducted?

We agree that a pulse-chase would be the way to definitively show this, however, we could never obtain sufficient number of DeltaRep2 sporozoites to do the chase. Clearly overall amounts of CSP were equivalent (see western blot in Fig 1B) but we could not tell from the western whether the CSP band for DeltaRep2 was cleaved or full-length because we only see one band in hemolymph sporozoites (we do not know why but this was also true of wildtype hemolymph sporozoites and could be the topic of another study!). The metabolic labeling experiment does help to answer this in that the majority of the label in a 45 minute pulse is full-length, with the 45 minute pulse being sufficient time to see the cleaved

form beginning to appear and it looks similar to the DeltaRep1 and Control lines. However we agree that this is not definitive and have modified the section as follows:

“No differences in CSP cleavage were observed in Δ Rep2 sporozoites (Supplementary Figure 4), suggesting that CSP is not prematurely cleaved. More definitive proof would be a pulse-chase experiment where we could measure the rate of cleavage, however, we could never harvest sufficient numbers of Δ Rep2 sporozoites to perform this experiment. Thus, current evidence suggests, but does not definitively prove, that the exposure of the TSR in Δ Rep2 sporozoites is not due premature CSP cleavage but rather to insufficient length of the mutant repeats to allow for N-terminal domain masking of the TSR. (beginning of page 7).

Dear Dr. Sinnis,

Thank you for submitting your revised manuscript (EMBOJ-2025-120089R) to The EMBO Journal for our consideration, and for your patience during peer review. Your manuscript was sent back to the original referees who had previously assessed the initial version of your work, and we have now received the full set of their comments, which are included below.

I am pleased to say that, as you will see, all three referees are very satisfied with the revision, recognize that this version of the manuscript is significantly improved and their initially raised concerns adequately addressed, and they now recommend publication of the manuscript with only few suggestions for minor corrections (including a few typos and the need for clarification on the used controls) that we would kindly ask you to address in a final version of your manuscript. Please note that referee #2 has provided a PDF copy of your manuscript with comments/edits, please find it attached and take it on board while revising your manuscript. Please also submit along with your manuscript a point-by-point response addressing the remaining comments and detailing any changes to the manuscript.

From the editorial side, there are also a few changes and corrections we need you to make in the final version of the manuscript before we can proceed with its formal acceptance and publication in The EMBO Journal:

- Please make sure that the co-corresponding author Amanda E. Balaban links a valid ORCID ID to the author's profile in our manuscript tracking system; this is mandatory for all corresponding authors. Please see our guide to authors for more information: <https://www.embopress.org/page/journal/14602075/authorguide#authorshipguidelines>.
- Please note that only the manuscript sections (e.g. Methods, Figure legends etc.) where the information can be found should be listed in the last column of the Author Checklist; the information itself should be included in the main manuscript file, not in the Author Checklist. Please also note that there are typos in the Checklist that should be corrected.
- The heading of the first page of the Appendix PDF file should be "Appendix for" followed by the manuscript's title and a brief Table of Contents including page numbers for the listed items. The nomenclature throughout the Appendix (and of the respective callouts in the main manuscript file) should be "Appendix Figure S#" and "Appendix Table S#".
- Materials and methods need to be described in the manuscript using our structured methods format, which is now required for all research articles. According to this format, the Methods section includes a single "Reagents and Tools Table" -listing key reagents, experimental models, software and relevant equipment including their sources and relevant identifiers- followed by a "Methods and Protocols" section describing the methods. Please download and fill our Reagents and Tools Table template (.docx), which you can find in our author guide: <https://www.embopress.org/page/journal/14602075/authorguide#structuredmethods>. When submitting your revised manuscript, please do not include the Reagents and Tools Table in the Methods section of the manuscript but instead upload it as a separate file choosing the file type "Reagent Table".
- Please note that EMBO press papers are accompanied online by:
 - A) a short (2 sentences) summary of the findings and their significance,
 - B) 2-5 short bullet points highlighting the key results, and
 - C) a synopsis image in .jpg or .png format that is exactly 550 pixels wide and 300-600 pixels high (the height is variable). Please note that the text needs to be legible at the final size.Please upload this information along with your revised manuscript (the text for A and B should be provided in a separate Word file).
- During our standard Figure checks we detected signs of potentially suboptimal processing of the gels shown in Appendix Figure S1B, which we need to investigate further. Please provide source data (in an "Appendix Figures" ZIP folder) for all gels (i.e. the original, uncropped and unedited gels, all available replicates) included in this Figure.
- During our routine data checks, our data editors have raised the following queries regarding figures, data, and legends. Please make sure that all requests below are completely addressed in the final version of your manuscript:
 1. Please provide the exact p values in the legends of Figures (or in the Figures) 1C, 2A, 3A, B; 5D, E; 6B, C; 7C, D, F, G; 8B, C, D, E.
 2. Please note that information related to "n" is missing in the legends of supplementary figure 4B.
 3. Please note that the error bars are not defined in the legends of supplementary figures 4B, 5B.
 4. Please note that the scale bar is missing for Figure 9B.
- The Movie files should be renamed to "Movie EV1-EV5" and the corresponding callouts should be updated accordingly. The legend of each Movie should be zipped together with the respective Movie file.
- The order of the manuscript sections must be corrected as follows: Title page - Abstract and Keywords - Introduction - Results

- Discussion - Methods - Data Availability - Acknowledgements - Disclosure and Competing Interests Statement - References - Figure Legends - main Tables (if there are any) - Expanded View Figure Legends.

Please also note that as part of the EMBO publications' Transparent Editorial Process, The EMBO Journal publishes online a Peer Review File along with each accepted manuscript. This File will be published in conjunction with your paper and will include the referee reports, your point-by-point response and all pertinent correspondence relating to the manuscript. You can opt out of this by letting the editorial office know (contact@embojournal.org). If you do opt out, the Peer Review File link will point to the following statement: "No Peer Review File is available with this article, as the authors have chosen not to make the review process public in this case."

We look forward to seeing a final version of your manuscript as soon as possible. Please let us know if you have any questions and use this link to submit your revision: <https://emboj.msubmit.net/cgi-bin/main.plex>.

Best regards,

Ioannis

Referee #1:

The authors have thoroughly revised their manuscript and have satisfactorily answered all of my questions. I thus fully endorse this manuscript for publication in this journal.

For the authors' information, I have two additional comments:

1. I found one more typo in the last sentence of the second but last paragraph of the introduction ("constrast" => "contrast"). This should be corrected.
2. I would like to share that we also see truncations in our recombinantly produced CSPs. Pure, recombinant PfCSP seems to be more sensitive compared to PbCSP and we observe truncations at the N-terminal end (confirmed by MS). Pure, recombinant PbCSP appears to be less sensitive as mentioned above, but it could well be that the truncations observed here for PbCSP obtained from native source are also caused by trimming of the N-terminus. This would also fit with the intrinsically disordered character of CSP's N-terminal domain, which renders this region more sensitive to proteolytic degradation (IDPs are notoriously sensitive to proteolytic clipping). The authors don't have to include this hypothesis into the manuscript, I simply wanted to share insights based on our experience.

I congratulate the authors with their work and thank the journal for considering me as a reviewer for this manuscript.

Referee #2:

This is a revised manuscript and the authors have paid close attention to the reviewers' original comments. I am happy with this revised version. I have made a few suggestions to the revised text that I think could improve the understanding of the results. The revision was a joy to read and the revised Figures are far more understandable.

Referee #3:

The authors have addressed the majority of my concerns and present a greatly revised and very interesting study. I am supportive of publication with the following minor requests:

1. I am comfortable with WT-GFP and RCon being used interchangeably as Controls however I request that the authors indicate which control was used specifically in each experiment in the figure legends or alternate location that facilitates ease of understanding. This is important for transparency and reproducibility.

2. The authors have responded to my query regarding amount of CSP in Rep2 based on metabolic labeling experiments. I noticed a typo, such that the revised sentence should be corrected: "...is not due 'to' premature CSP cleavage."

Response to Reviewer and Editor Comments

Referee #1:

The authors have thoroughly revised their manuscript and have satisfactorily answered all of my questions. I thus fully endorse this manuscript for publication in this journal.

For the authors' information, I have two additional comments:

1. I found one more typo in the last sentence of the second but last paragraph of the introduction ("constrast" => "contrast"). This should be corrected.

Thank-you and corrected!

2. I would like to share that we also see truncations in our recombinantly produced CSPs. Pure, recombinant PfCSP seems to be more sensitive compared to PbCSP and we observe truncations at the N-terminal end (confirmed by MS). Pure, recombinant PbCSP appears to be less sensitive as mentioned above, but it could well be that the truncations observed here for PbCSP obtained from native source are also caused by trimming of the N-terminus. This would also fit with the intrinsically disordered character of CSP's N-terminal domain, which renders this region more sensitive to proteolytic degradation (IDPs are notoriously sensitive to proteolytic clipping). The authors don't have to include this hypothesis into the manuscript, I simply wanted to share insights based on our experience.

Thank-you for sharing this. We do not work much with recombinant CSP but are currently investigating the N-terminus of native CSP and I agree with you that the disordered region of the N-terminus is highly sensitive to proteolytic clipping.

I congratulate the authors with their work and thank the journal for considering me as a reviewer for this manuscript.

We appreciated your thoughtful review which made this a much better paper.

Referee #2:

This is a revised manuscript and the authors have paid close attention to the reviewers' original comments. I am happy with this revised version. I have made a few suggestions to the revised text that I think could improve the understanding of the results. The revision was a joy to read and the revised Figures are far more understandable.

Thank-you for this feedback! We appreciate your attention to detail which has made this a better manuscript.

Regarding the additional editorial changes in the pdf, it seems that you used the original pdf and that the original comments were marked as "redacted" and in purple with the new comments in red. However since this was the original version of the manuscript, some of the comments in red have been addressed through rewording of the text in response to Reviewers 1 and 3. Also my version of Acrobat did not show me the new comment in red in the Significance Statement so I could not change this. This is indicated with a note in the text of the 2nd revision. In all other cases we have changed the manuscript as you outline and again, the changes were a definite positive!

Referee #3:

The authors have addressed the majority of my concerns and present a greatly revised and very interesting study. I am supportive of publication with the following minor requests:

1. I am comfortable with WT-GFP and RCon being used interchangeably as Controls however I request that the authors indicate which control was used specifically in each experiment in the figure legends or alternate location that facilitates ease of understanding. This is important for transparency and reproducibility.

We've now went back through our lab notebooks and figured out how many of the biological replicates were performed with WT-GFP and how many with RCon and added this to the Source Data excel file for each Figure. Please note that the raw data from Figure 4 was used for all of the analyses in Figure 5 so this figure did not require updating. Also, Figures 8 and 9 did not use RCon or WT-GFP parasites so their raw data also did not require updating.

2. The authors have responded to my query regarding amount of CSP in Rep2 based on metabolic labeling experiments. I noticed a typo, such that the revised sentence should be corrected: "...is not due 'to' premature CSP cleavage."

Now corrected

From the editorial side, there are also a few changes and corrections we need you to make in the final version of the manuscript before we can proceed with its formal acceptance and publication in The EMBO Journal:

- Please make sure that the co-corresponding author Amanda E. Balaban links a valid ORCID ID to the author's profile in our manuscript tracking system; this is mandatory for all corresponding authors. Please see our guide to authors for more information:

<https://www.embopress.org/page/journal/14602075/authorguide#authorshipguidelines>.

While Dr. Balaban's orcid ID is 0000-0002-1072-9140, she could not link it to her EMBO profile and the staff could not do this on her behalf because of 'security' concerns. We have therefore removed her as co-corresponding author.

I have also had trouble with Dr. Ha's contact information on the web platform. The title page of the manuscript is correct and even though Dr. Ha succeeded in linking his orcid ID to the EMBO paper, his contact information on your website was not updated. Just wanted to let you know.

- Please note that only the manuscript sections (e.g. Methods, Figure legends etc.) where the information can be found should be listed in the last column of the Author Checklist; the information itself should be included in the main manuscript file, not in the Author Checklist. Please also note that there are typos in the Checklist that should be corrected.

Typo corrected and now only the manuscript section is listed in the last column.

- The heading of the first page of the Appendix PDF file should be "Appendix for" followed by the manuscript's title and a brief Table of Contents including page numbers for the listed items. The nomenclature throughout the Appendix (and of the respective callouts in the main manuscript file) should be "Appendix Figure S#" and "Appendix Table S#".

Done

- Materials and methods need to be described in the manuscript using our structured methods format, which is now required for all research articles. According to this format, the Methods section includes a single "Reagents and Tools Table" -listing key reagents, experimental models, software and

relevant equipment including their sources and relevant identifiers- followed by a "Methods and Protocols" section describing the methods. Please download and fill our Reagents and Tools Table template (.docx), which you can find in our author guide: <https://www.embopress.org/page/journal/14602075/authorguide#structuredmethods>. When submitting your revised manuscript, please do not include the Reagents and Tools Table in the Methods section of the manuscript but instead upload it as a separate file choosing the file type "Reagent Table".

Reagent Table now included in the uploads.

- Please note that EMBO press papers are accompanied online by:

A) a short (2 sentences) summary of the findings and their significance,

B) 2-5 short bullet points highlighting the key results, and

C) a synopsis image in .jpg or .png format that is exactly 550 pixels wide and 300-600 pixels high (the height is variable). Please note that the text needs to be legible at the final size.

Please upload this information along with your revised manuscript (the text for A and B should be provided in a separate Word file).

Summary and bullet points now provided in a separate word document.

Synopsis image provided.

- During our standard Figure checks we detected signs of potentially suboptimal processing of the gels shown in Appendix Figure S1B, which we need to investigate further. Please provide source data (in an "Appendix Figures" ZIP folder) for all gels (i.e. the original, uncropped and unedited gels, all available replicates) included in this Figure.

We have now uploaded these files as "Appendix Figures Source Data"

- During our routine data checks, our data editors have raised the following queries regarding figures, data, and legends. Please make sure that all requests below are completely addressed in the final version of your manuscript:

1. Please provide the exact p values in the legends of Figures (or in the Figures)

When $p < 0.0001$, our statistical analysis software does not give an exact p-value and in these cases there were no changes made to p-values in the legends. It is also important to note that because these statistical tests were initially performed 3 years ago, and Prism, which is the program we use for our statistical analyses has been 'updated', in some cases the same tests now gave slightly different results. In these instances we changed the corresponding figure and have uploaded revised figures.

Specifics for each request:

1C – Redoing the statistical analyses on these data led to a revision of one of the p-values to $p < 0.0001$. The figure and legend have been changed accordingly.

2A – exact p-value for one of the comparisons is now included and the others were < 0.0001 .

3A - all p-values were < 0.0001

3B – exact p-value now added

5D, E – The one-way Anova changed to $p < 0.0001$ and exact p-values are now included where appropriate.

6B, C - all of the original values were $p < 0.0001$ but on redoing the stats, adhesion site turnover in the Control vs Rep 2 is now statistically significant, $p = 0.0126$. The figure and corresponding text in the manuscript have been updated.

7C, D, F, G – Now added exact p-values for 7D, and the significance of one of the values went from ** to * with the new version of Prism so that figure has also been updated. For 7C, F, and G, all $p < 0.0001$.

8B, C, D, E – no p-values were in this figure/figure legend.

2. Please note that information related to "n" is missing in the legends of supplementary figure 4B. Now added to the figure legend: "Densitometry values for Control samples are combined from the two blots in panel A (n=2 with mean +/- standard deviation) and for Δ Rep1 and Δ Rep2 are from the one sample shown in panel A (n=1)."

3. Please note that the error bars are not defined in the legends of supplementary figures 4B, 5B. See above for 4B and for 5B now added "Shown are the mean +/- standard deviation"

4. Please note that the scale bar is missing for Figure 9B.

It must have gotten lost when I generated the high resolution figures – now added back!

- The Movie files should be renamed to "Movie EV1-EV5" and the corresponding callouts should be updated accordingly. The legend of each Movie should be zipped together with the respective Movie file.

Done – please note that there were no legends for the movies as the title of the movie is sufficient. We had 3 very thorough reviewers and they had no problem understanding the movies.

- The order of the manuscript sections must be corrected as follows: Title page - Abstract and Keywords - Introduction - Results - Discussion - Methods - Data Availability - Acknowledgements - Disclosure and Competing Interests Statement - References - Figure Legends - main Tables (if there are any) - Expanded View Figure Legends.

Done

Please also note that as part of the EMBO publications' Transparent Editorial Process, The EMBO Journal publishes online a Peer Review File along with each accepted manuscript. This File will be published in conjunction with your paper and will include the referee reports, your point-by-point response and all pertinent correspondence relating to the manuscript. You can opt out of this by letting the editorial office know (contact@embojournal.org). If you do opt out, the Peer Review File link will point to the following statement: "No Peer Review File is available with this article, as the authors have chosen not to make the review process public in this case."

Its fine by me

Dear Photini,

Congratulations on an excellent manuscript! I am very pleased to inform you that it has been accepted for publication in The EMBO Journal. Thank you for comprehensively addressing the initially raised referee concerns and the editorial requests for corrections and changes.

There is only one minor correction we still need you to make in the Appendix file: in Appendix Figure S4, statistics (mean and standard deviation) has been calculated for $n=2$ (control), which is not allowed according to our journal's policy. When $n=2$, no statistics can be calculated and shown; instead, the individual data points should be shown. I would be grateful if you could please revise this Figure (and its legend, as appropriate) and send me the new Appendix PDF file by e-mail.

Your manuscript will then be processed for publication by EMBO Press. It will be copy edited and you will receive page proofs prior to publication. Please note that you will be contacted by Springer Nature Author Services to complete licensing and payment information.

If you have any questions, please do not hesitate to contact the Editorial Office. Thank you for your contribution to The EMBO Journal. Working with you has been a pleasure.

Best regards,

Ioannis
